# Importance of size representation and morphology in modelling optical properties of black carbon: comparison between laboratory measurements and model simulations

Baseerat Romshoo[1], Mira Pöhlker[1], Alfred Wiedensohler[1], Sascha Pfeifer[1], Jorge Saturno[2], Andreas Nowak[2], Krzysztof Ciupek[3], Paul Quincey[3], Konstantina Vasilatou[4], Michaela N. Ess[4], Maria Gini[5], Konstantinos Eleftheriadis[5], Chris Robins[3], François Gaie-Levrel[6] and Thomas Müller[1]

[1]Leibniz Institute for Tropospheric Research, 04318, Leipzig, Germany
[2]Physikalisch-Technische Bundesanstalt (PTB), Braunschweig, 38116, Germany
[3]Atmospheric Environmental Science Department, National Physical Laboratory (NPL), Teddington, TW11 0LW, UK
[4]Federal Institute of Metrology METAS, Bern-Wabern, 3003, Switzerland
[5]Environmental Radioactivity Laboratory, Institute of Nuclear & Radiological Sciences and Technology, Energy & Safety (INRASTES), N.C.S.R. Demokritos, Attiki, 15310, Greece
[6]Laboratoire National de Métrologie et d'Essais (LNE), Paris, 75015, France

*Correspondence to*: Baseerat Romshoo (baseerat@tropos.de

## Abstract

Black carbon (BC) from incomplete combustion of biomass or fossil fuels is the strongest absorbing aerosol component in the atmosphere. Optical properties of BC are essential in climate models for quantification of their impact on radiative forcing. The global climate models, however, consider BC to be spherical particles which causes uncertainties in their optical properties. Based on this, an increasing number of model-based studies provide databases and parametrization schemes for the optical properties of BC using more realistic fractal aggregate morphologies. In this study, the reliability of the different modelling techniques of BC were investigated by comparing them to laboratory measurements. In the first step, the modelling techniques were examined for bare BC particles, and in the second step, for BC particles with organic material. A total of six morphological representations of BC particles were compared, three each for spherical and fractal aggregate morphologies. In general, the aggregate representation performed well for modelling the particle light absorption coefficient $\sigma_{abs}$, single scattering albedo SSA, and mass absorption cross-section $MAC_{BC}$ for laboratory generated BC particles with volume mean mobility diameters $d_{p,V}$ larger than 100 nm. However, for modelling Ångström absorption exponent AAE it was difficult to suggest a method due to size-dependence, although the spherical assumption was in better agreement in some cases. The BC fractal aggregates are usually modelled using monodispersed particles since their optical simulations are computationally expensive. In such studies, the modelled optical properties showed a 25% uncertainty in using the monodisperse size method. It is shown that using the polydisperse size distribution in combination with fractal aggregate morphology reduces the uncertainty in measured $\sigma_{abs}$ to 10%, for particles with $d_{p,V}$ between 60-160 nm.
Furthermore, the sensitivities of the BC optical properties to the various model input parameters such as the real and imaginary parts of the refractive index ($m_{re}$ and $m_{im}$), the fractal dimension ($D_f$), and the primary particle radius ($a_{pp}$) of an aggregate were investigated. When the BC particle is small and rather fresh, the change in the $D_f$ had relatively little effect on the optical properties. There was, however, a significant relationship between $a_{pp}$ and the particle light scattering, which increased by a factor of up to six with increasing total particle size. The modelled optical properties of BC are well aligned with laboratory-measured values when the following assumptions are used in the fractal aggregate representation: $m_{re}$ between 1.6 to 2, $m_{im}$ between 0.50 to 1, $D_f$ from 1.7 to 1.9, and $a_{pp}$ between 10 to 14 nm. Overall, this study provides experimental support for emphasizing the importance of an appropriate size representation (polydisperse size method) and an appropriate morphological representation for optical modelling and parametrization scheme development of BC.

## Introduction

Soot particles are produced by incomplete combustion of carbonaceous materials such as fossil fuels, biomass, and biofuels. Black carbon (BC), a major component of soot and also known as light-absorbing carbon, contributes significantly to global warming along with $CO_2$, methane, and volatile organic compounds (VOCs) (IPCC, 2021).

On a regional scale, black carbon can significantly perturb the climate (Wang, 2004; Menon et al., 2002). In developing areas such as China, South Asia, and South East Asia, rapid urbanization has caused an alarming increase in the BC mass fraction of the total particle mass concentration (Wiedensohler et al., 2018b; Madueño et al., 2019). Moreover, increasing mass concentrations of BC are degrading air quality and causing adverse effects on human health (Pöschl., 2005; Janssen et al., 2011).

High-resolution transmission electron microscopy (TEM) analysis of BC samples from ambient and laboratory studies revealed that BC particles comprise agglomerates made from numerous graphitic spherules (Betrancourt et al., 2017; Gini et al., 2016). Over time, BC agglomerates undergo complex changes in their size, morphology, and composition, depending on post-emission atmospheric conditions (Fierce et al., 2015). TEM images from Shanghai's atmosphere presented by Fu et al. (2012) showed a variety of BC-containing particles at various stages of aging, of which some semi-aged particles retained fractal aggregate morphology. The BC particles are often found together with other combustion by-products such as organic matter, which enhance the particle light absorption through the lensing effect (Fuller et al., 1999). With increasing residence time of BC in the atmosphere, an aging process occurs, leading to a growth of BC agglomerates into much more compact structures. There are several reasons for this, including the formation of coatings and hygroscopic effects (Petzold et al., 2005; Bond et al., 2006; Abel et al., 2003). Cloud processing such as water condensation or evaporation also restructures the BC particles into more compact shapes (Bhandari et al., 2019).

The impact of BC particles on climate is studied by estimating their radiative forcing properties using global climate models (IPCC 2021; Krüger et al., 2022; Jacobson., 2001). In order to simulate the BC radiative forcing in global models, the estimates of various BC optical properties, such as particle light scattering, and mass absorption cross-sections, must be taken into account (Bond et al., 2013; Ciupek et al., 2021). The morphological structure of BC particle plays an important role in determining their light scattering and absorption coefficients(He et al., 2015). The Lorentz-Mie theory (Mie, 1908) is often used to calculate the optical properties of BC particles (Bohren and Huffman, 1998; Bond et al., 2013). This theory is preferred because of the computational simplicity and wide applicability. However, studies have shown large discrepancies in the results of Lorentz-Mie theory when compared with ambient measurements (Adachi et al., 2010; Wu et al., 2018). Moreover, given the complex aging process of BC agglomerates, it is unrealistic to assume BC particles as spherical particles.

Due to the limitations of the Lorentz-Mie theory, the number of studies on the computation of BC optical properties assuming a fractal aggregate morphology has increased (e.g., Berry and Percival, 1986; Kahnert and Kanngießer, 2020; Smith and Grainger, 2014; Liu et al., 2018). To model the optical properties of such fractal BC aggregates, the Rayleigh-Debye-Gans (RDG) approximation (Sorensen, 2011), the discrete dipole approximation DDA (Purcell and Pennypacker, 1973), and the T-matrix method (Mackowski and Mishchenko, 2011) have been used (Adaichi et al., 2010; Kahnert, 2010; Li et al., 2016; Scarnato et al., 2013). Parametrization schemes and databases for the optical properties of BC as fractal aggregates have been developed, and proposed for applications in climate models by Smith and Grainger (2014), Romshoo et al., (2021), Liu et al., (2019), and Luo et al., (2018a).

In addition to the various numerical studies conducted on the optical properties of BC aggregates, there is a scientific need to examine the reliability of the modelling techniques, and their comparability with actual measurements. Liu et al., 2018 provided a theoretical overview of how sensitive the radiative properties of black carbon is to their complex morphologies. The geometric-optics surface wave (GOS) approach was used to calculate the BC light scattering properties at different aging stages and compare them with the measured values (He at el., 2015). Forestieri et al. (2018) measured and modelled the mass absorption cross-sections ($MAC_{BC}$) for bare flame-generated black carbon. Due to the high computational time of optical simulations, most of the modelling studies are limited to monodisperse particles (Kahnert, 2010; Adaichi et al., 2010; Kahnert and Kanngießer, 2020; Smith and Grainger, 2014; Romshoo et al., 2021; Liu et al., 2019; Luo et al., 2018a). However, for atmospheric applications, ensemble-averaged optical properties for given particle number size distributions are needed (Bond et al., 2013). Therefore, it would be reasonable to investigate the performance of different modelling approaches for calculating the ensemble-averaged optical properties.

In order to model BC as an aggregate morphology, it is necessary to be aware that BC aggregates are composed of tiny spherules called 'primary particles' or 'monomers' (Tian et al., 2006; Betrancourt et al., 2017). TEM images show that these primary particles measure between 10 and 30 nm in diameter, depending on the source of combustion, and the interaction among the various mechanisms involved in black carbon formation (Kholghy et al. 2013; Park et al. 2005). The morphology of the BC aggregates is described by a parameter called fractal dimension $D_f$ (Köylü et al., 1995). Depending on the dynamics of the collisions, and the restructuring and condensation of organic matter present in the atmosphere after emission, the $D_f$ of black carbon can vary from 1.5 up to ~ 2.8 (Wentzel et al., 2003; Gwaze et al., 2006; Ghazi et al., 2013). The size of the BC primary particle and the fractal dimension are important parameters used in optical modelling studies. However, it is unclear to what extent the assumptions of these input parameters are important when compared to ambient or laboratory measurements.

In this work, we examine modelling methods of BC optical properties for both monodisperse as well as polydisperse aerosol particles. The novelty of this study is the improvement of the modelling techniques for optical

properties of BC in order to match their equivalent laboratory measurements. The study is structured as follows. An overview of the laboratory methods is given first, followed by the discussion of the various aspects of modelling the optical properties of BC, such as their representation, selection of the particle sizes, various model input parameters, and the optical model itself. Furthermore, the modelling techniques for two kinds of BC particles are investigated. We begin with modelling the first kind i.e., bare BC particles, evaluating the assumptions of various modelling parameters (for e.g., $m_{re}$, $m_{im}$, $D_f$, and $a_{pp}$) and comparing them to experimental results. The modelling techniques for the second kind i.e., BC particles containing organics is discussed next. Finally, a summary and recommendations for future modelling studies are provided.

## 2    Methods

### 2.1    Laboratory generated black carbon

The measurements reported in this study were from two laboratory campaigns for characterization of black carbon. Experiment E1 involved measurements of thermally denuded nascent black carbon particles conducted at the National Meteorology Institute of Germany (Physikalisch Technische Bundesanstalt, Braunschweig). In the second experiment (E2), measurements of untreated nascent black carbon particles were performed at the Leibniz Institute for Tropospheric Research.

### 2.1.1    Generation of black carbon particles

For this study, three different mini-CAST soot generators  (Jing Ltd, Switzerland) were used, which can generate black carbon particles within a wide range of concentrations, sizes, and chemical compositions (Moore et al. 2014; Ess et al., 2019). Mini-CAST  soot generators are diffusion-based or premixed flame-based, which generate black carbon particles after combustion with a mixture of fuel (propane) and air (Jing et.al, 2014). In the diffusion flame based mini-CAST, propane is mixed with oxidation air at the flame via diffusion, using nitrogen for quenching the flame. In the premixed version of mini-CAST propane and air are mixed before being injected into the flame which results in a premixed (or partially premixed) flame. Depending on the flame type, either of these mini-CASTs can control the black carbon characteristics by varying the flow rates of fuel, oxidation air, and nitrogen. A key parameter describing the operating conditions of mini-CAST is the overall fuel-to-air ratio, also called the flame equivalence ratio, $\phi$. The generator can be operated in a fuel-rich condition when $\phi > 1$, whereas fuel-lean (or near-stoichiometric) condition is defined by $\phi < 1$. Moore et al., 2014 mapped the operation of the soot generator mini-CAST 4202 (Zollikofen BE, Switzerland; Jing 1999) for a wide range of operating conditions, providing an optimal guide for laboratory-based black carbon generation using a mini-CAST burner. In this study, a total of four mini-CASTs were used with different operating conditions during both laboratory campaigns. The mini-CASTs were operated at fuel-lean operating conditions with flame equivalence ratios ranging from 0.74 to 1.01, producing black carbon particles with volume mean mobility diameter ($d_{p,\bar{V}}$) between 53 and 182 nm. Table 1 provides an overview of the operating conditions of the mini-CASTs for both E1 and E2.

**Table 1.** Details of the different cases in experiments E1 and E2: the operating conditions and resulting properties of the particles such as the mobility diameters ($d_{p,\bar{N}}$ and $d_{p,\bar{V}}$), ratio of the elemental to total carbon (EC/TC), and single scattering albedo (SSA) at wavelength of 660 nm. All the mentioned properties will be defined in the next sections.

| Experiment series | Case | Mini-CAST model | Propane (mlpm) | N2/ Mixing air* (lpm) | Oxidation air (lpm) | $\phi$ | $d_{p,\bar{N}}$ | $d_{p,\bar{V}}$ | EC/ TC | SSA |
|---|---|---|---|---|---|---|---|---|---|---|
| E1 | I | MC 5203C | 140 | 0.61 | 3.30 | 1.01 | 38 | 60 | - | 0.014 |
| E1 | II | MC 5203C | 140 | 0.56 | 3.60 | 0.93 | 71 | 106 | - | 0.024 |
| E1 | III | MC 5203C | 140 | 0.33 | 3.30 | 1.01 | 105 | 160 | - | 0.074 |
| E1 | IV | MC 5203C | 84 | 0.00 | 2.72 | 0.74 | 105 | 160 | - | 0.042 |
| E2 | V | MC 5201BC | 60 | 0.42 | 1.10 | 0.94 | 56 | 83 | 0.35 | 0.011 |
| E2 | VI | MC 5201BC | 60 | 0.39 | 1.10 | 0.96 | 89 | 126 | 0.69 | 0.053 |
| E2 | VII | MC 5201BC | 60 | 0.23 | 1.30 | 0.94 | 129 | 181 | 0.68 | 0.062 |
| E2 | VIII | MC 5203C | 140 | 0.56 | 3.60 | 0.93 | 48 | 86 | 0.35 | 0.054 |
| E2 | IX | MC 5203C | 140 | 0.00 | 3.30 | 1.01 | 122 | 174 | 0.66 | 0.112 |
| E2 | X | MC 5303C | 140 | 0.30 | 4.20 | 0.80 | 84 | 122 | 0.68 | 0.045 |
| E2 | XI | MC 5303C | 140 | 0.00 | 4.20 | 0.80 | 122 | 181 | 0.62 | 0.083 |

*For mini-CAST 5201BC

### 2.1.2 Objectives of laboratory experiment E1 and E2

Experiment E1: The objective this experiment was to obtain the size, and the optical properties of black carbon particles after removal of the volatile organic content, which are expected to represent bare black carbon particles as closely as possible. Figure A1 shows a schematic of the experimental setup used in experiment E1. The black carbon particles were produced with a mini-CAST 5203 Type C. The mini-CAST 5203C consists of three diffusion flames, generating black carbon particles under fuel-lean operating conditions. The aerosols generated from mini-CAST 5203C were passed through a Catalytic Stripper (Catalytic Stripper Model CS015, Catalytic Instruments, Rosenheim, Germany) to remove the volatile contents, in this case, mainly organic carbon. For each case in E1 (Table. 1), the Catalytic Stripper was operated at unheated condition, at 150°C condition (BC particles pass through the Catalytic Stripper operated at 150°C), and at 350°C condition (BC particles pass through the Catalytic Stripper at 350°C). Particles coming out of the Catalytic Stripper are then passed through several instruments that measure particle number size distribution, particle light extinction, absorption, and scattering. Detailed information about these measurements is provided in Appendix A.

Experiment E2: In this experiment, the size, the composition, and the optical properties of untreated nascent black carbon particles produced by the different mini-CAST burners at different operating conditions were measured. The schematic diagram of the experimental setup used in E2 is shown in Figure A2. Three mini-CAST models were used in this experiment including a mini-CAST 5203 Type C, a mini-CAST 5201 Type BC, and a mini-CAST 5303 Type C were used. The mini-CAST 5201BC burner was operated in the partially premixed flame mode (Ess et al. 2019, Ess et al. 2021). The flow settings of propane, nitrogen or mixing air (mini-CAST 5201 BC), and oxidation air were adjusted in order to obtain black carbon particles of specific size, as shown in Table 1 by the corresponding number mean mobility diameter ($d_{p,\bar{N}}$), and the volume mean mobility diameter ($d_{p,\bar{v}}$). The details of the flow settings for the three mini-CAST models used are shown in Table 1. The particles generated from the black carbon generators are delivered to various instruments to measure their number size distributions, aerosol mass concentration, chemical composition, particle light extinction, absorption, and scattering coefficients. The details about the instrumentation used are shown in Appendix A.

## 2.2 Fundamentals of modelling optical properties of black carbon particles

### 2.2.1 Morphology of black carbon and representations for modelling

In order to model the optical properties of black carbon, it is important to choose the most appropriate morphological representation for black carbon particle. This step is considered particularly important because the modelled optical properties were further validated with the measurements from E1 and E2. TEM images were not available for this study, therefore, the morphological representations of black carbon were selected based on TEM images from a previous laboratory study using the mini-CAST generators (Ess et al., 2021; Ouf et al., 2016). In the mini-CAST generator, BC particles produced have fractal morphologies, with varying amounts of organics attached to the edges, without altering the inner structure of the core (Ouf et al., 2016). In addition to the TEM images from Ess et al. (2021), the operating conditions of the mini-CAST burners during experiments E1 and E2 (Table. 1), and the fraction of organic carbon of BC particles from E2 were also considered while selecting the morphological representations. A more detailed discussion of the TEM images of BC particles taken from mini-CAST and atmosphere can be found in the Supplementary material.

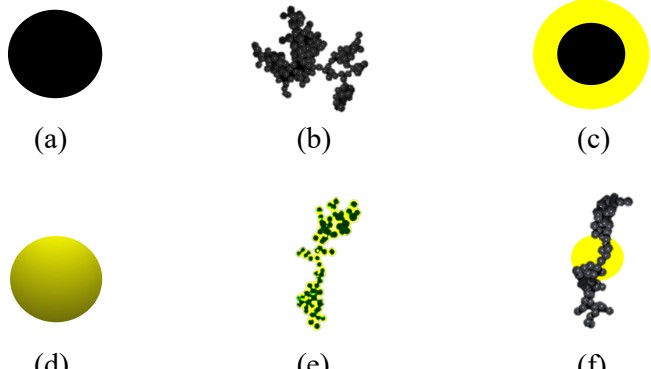

**Figure 1.** Morphological representations of black carbon used in this study: **(a) sphere, (b) aggregate, (c) coated sphere, (d) homogeneously mixed sphere, (e) coated aggregate , and (f) aggregate partly enclosed in sphere.**

For modelling the particles from the denuding experiment E1, the simulated particles are assumed to be bare black carbon, since a Catalytic Stripper was used to remove the volatile organic matter. Some residuals, however, are left behind by the Catalytic Strippers, depending on the volatility of the organic matter. Mamakos et al. (2013) reported that in the 21–250◦C temperature range, the Catalytic Stripper is able to remove up to 96% of the more volatile fraction of organic matter. However, in the 250–500◦C temperature range, the Catalytic Stripper removes 30–60% of the less volatile organic matter. . This must be noted when comparing the modelled optical results with their equivalent laboratory measurements.

Two morphological representations of bare BC particles were used as shown in Figure 1(a, b). The first one is a sphere(Fig. 1a); the second one is a fractal aggregate(Fig. 1b). The 'sphere' representation is the most simplified representation used by fellow researchers (Bond et al., 2013). An 'aggregate' representation shows the realistic morphology of the BC aerosols when they are formed by combustion (Michelsen et al., 2017; Ess et al., 2021). The morphology of such fractal aggregates is mathematically described by Eggersdorfer et al., (2012):

$$N_{\text{pp}} = k_{\text{fm}} \left( \frac{D_{\text{p}}}{2a_{\text{pp}}} \right)^{D_{\text{fm}}} , \qquad\qquad (1)$$

where, $a_{\text{pp}}$ is the radius of primary particles, $N_{\text{pp}}$ is the number of primary particles, $D_{\text{fm}}$ is the mass-mobility exponent, $D_{\text{p}}$ is the mobility diameter, and $k_{\text{f}}$ is a dimensionless pre-factor.

In the experiment E2, additional information about the chemical composition of the black carbon particles were available from the EC/OC analysis conducted on the loaded quartz filters. Based on the EC/OC analysis results, the various morphological representations of BC particles containing organics are simulated. Four models for BC particles containing organics were used to represent the particles generated from E2. The four representations are shown in Figure 1 (c) to (f) for coated spheres, homogeneously mixed spheres, coated aggregates and aggregates partly enclosed in sphere. The 'coated sphere' comprised of an inner spherical BC core enclosed within a shell of organic carbon. In the 'homogeneously mixed sphere', BC and organic carbon were internally mixed following the volume mixing rule (Chylek et al., 1995) to form a homogenized mixture. The 'coated sphere' and 'homogeneously mixed sphere' are the simplified models to represent coated BC aerosols. The 'coated aggregate' is a realistic representation, morphologically similar to the 'aggregate' (Figure 1b), with the difference that each monomer is coated with a layer of organic carbon. Ouf et al. (2016) conducted NEXAFS analysis on BC produced from a diffusion flame-based mini-CAST burner and found that organics (by-products of the combustion) get attached to the edge of graphite crystallites without changing the inner structure of the core. This laboratory result can be simulated for coated BC in radiative modelling studies by assuming a spherical coating around each individual primary particle of a BC aggregate (Luo et al., 2018b). This method was used to simulated 'coated aggregate' representation in our study. BC particles that we modelled in our study have a fraction of organics ($f_{\text{oc}}$) up to 53%, therefore they are assumed to have a less compact and chain-like structure. In such cases, where the BC aggregate does not have a completely compact structure, the results using the 'coated aggregate' representation are expected to be reliable (Luo et al., 2018b). Moreover, Kahnert et al., 2017 compared the coating model (closed-cell model) used in this study to a realistic model, which showed good comparability. Finally, the 'aggregates partly enclosed in sphere' represented a model for aged black carbon, comprising of an 'aggregate' (Figure 1b) immersed in a sphere of organic carbon. Since this study simulates laboratory-produced black carbon, the particles are not likely to resemble those in the 'aggregates partly enclosed in sphere' representation. It was

nevertheless included in the study for the sake of comparison. Further details of how the six morphological representations shown in Fig. 1 were modelled will be explained in the following sections.

### 2.2.2 Construction rules for spherical particles

In the 'sphere' and 'homogeneously mixed sphere' representation, the diameters of the spheres were taken from the SMPS size distributions obtained from the laboratory experiments. The 'coated sphere' representation consisted of two spheres; the diameter of the outer sphere ($D_o$) was directly taken from the SMPS size distributions. The diameter of the inner sphere ($D_i$) was obtained by:

$$D_i^3 = (1 - f_{oc}) D_o^3, \tag{2}$$

where $f_{OC}$ is the fraction of organic carbon obtained from the results of EC/OC analysis as:

$$f_{oc} = 1 - \frac{EC}{TC} = 1 - \frac{EC}{EC+OC} \tag{3}$$

where $\frac{EC}{TC}$ is the volume ratio of elemental carbon to the total carbon ($TC = OC + EC$). The volume ratio is derived from the EC/OC analysis after dividing the masses by their respective densities. In this study, it was assumed that elemental carbon corresponds to black carbon. The density of elemental carbon $\rho_{EC}$ was taken as 1.8 g cm$^{-3}$ (Park et al., 2004), and the density of organic carbon as 1.1 g cm$^{-3}$ (Schkolnik et al., 2007).

### 2.2.3 Construction rules for aggregate particles

For simulating the 'aggregate', 'coated aggregate', and 'aggregate and sphere' representations, the number of primary particles $N_{pp}$ per aggregate, and the radius of primary particle $a_{pp}$ must be determined. In the previous studies about the comparison of modelled and measured optical properties of soot aggregates, the $N_{pp}$ was determined by dividing the measured mass of total particle by the estimated mass of a spherule (Forestieri et al., 2018) or reconstructed using results from TEM analysis (He et al., 2015). In our study, we investigated the methods for estimating the $N_{pp}$ in absence of mass or TEM results. Three different conversion methods for calculating the number of primary particles $N_{pp}$ per aggregate were applied in this study. In the first method by Rissler et al. 2012, the particle mass estimated using the $\rho_{eff}$ is divided by the estimated mass of a spherule. The second technique described by Sorensen. (2011) uses the mobility mass scaling exponent in conjunction with the concept that black carbon aggregates fall into the slip regime. The third method, developed by Schmidt-Ott. (1988) is based on a power law function. Further details of the three methods for calculation of $N_{pp}$ are provided in Appendix B.

The radius of primary particle $a_{pp}$ is used in all the three methods for calculating the number of primary particles $N_{pp}$ per aggregate. Diffusion flame-based generators like the mini-CAST burners, produce BC aggregates with primary particle radius (app) between 4 and 14 nm (Bourrous et al., 2018; Mamakos et al., 2013 ). Kahnert (2010) pointed out the insensitivity in the optical properties when the radii of the primary particle fall in the range of 10–25 nm. Due to absence of measurements for $a_{pp}$, and for the sake of simplicity, a constant average value of $a_{pp}$ = 14 nm was used for the entire study, except for the part of sensitivity analysis discussed in the next section.    In the 'coated aggregate' representation, a layer of organic carbon was present around each primary particle comprising the BC aggregate. Following equation (2), the relationship between the fraction of organic carbon ($f_{oc}$), the outer radius of the primary particle ($a_o$) and the inner radius of the primary particle ($a_{in}$) were determined. It must be noted that in the 'coated aggregate' representation, the primary particles of the aggregates generated from the Diffusion Limited Aggregation (DLA) software have a radius equal to $a_o$. In the next step, a smaller sphere with a radius of $a_{in}$ is placed at the center of the primary particle representing the BC core. In the 'aggregate partly enclosed in sphere' representation, after generating an aggregate comprising of black carbon, a sphere of organic carbon was placed at the center of mass of the black carbon aggregate. The radius of the sphere of organic carbon ($R_{so}$) is obtained by:

$$R_{so}^3 = f_{oc}(a_{app}^3 \cdot N_{pp}), \tag{4}$$

When a sphere of organic carbon is placed around parts of BC aggregate in the 'aggregate partly enclosed in sphere' representation, the parts of black carbon aggregate inside the sphere reduces the volume of organic carbon. Iteratively increasing the radius of the sphere of organic carbon would replace this lost volume. In this study, since non-compact aggregates were used and the fraction of organics ($f_{oc}$) was up to 53%,only a small portion of BC aggregate was present inside the organic sphere. A sensitivity study was conducted to test how the absorption

cross-section changes when the radius of the sphere of organic carbon is iteratively increased. The results of this sensitivity analysis showed that the absorption cross-section varied by 2 to 3% after iteratively increasing the radius of the organic carbon sphere. Thus, for the sake of simplicity, the particles were left as they are. However, when modelling coated aggregates with more compact structures or high coating fractions, it is recommended to apply the iteration schemes to each particle.

### 2.2.4 Other parameters from literature

Simulation of optical quantities with scattering calculations requires a number of assumptions about the morphology of the particles and the refractive indices. In the first experiment, E1, the composition of the simulated morphological representations 'sphere', and 'aggregate' was assumed to be bare black carbon, i.e., elemental carbon in nature. The real and imaginary parts of the refractive index, $m_{re}$ and $m_{im}$, respectively, were taken from a study by Kim et al., 2015. The values of $m_{re}$ and $m_{im}$ for EC at wavelengths of 467, 530, and 660 nm are summarized in Table A2. The refractive index of the OC in experiment E2 is also taken from Kim et al., 2015, for the representations of 'coated sphere', 'coated aggregate', and 'aggregate partially enclosed in sphere'. However, for the 'homogeneously mixed sphere', the effective complex refractive index m was calculated from the volume-mixing rule (Chylek et al., 1995). The values of $m_{re}$ and $m_{im}$ for OC used in this study are summarized in Table A2.

In the 'aggregate', 'coated aggregate', and 'aggregate partially enclosed in sphere' representation, the morphology of the particle is described by the fractal dimension $D_f$. The rrepresentative values for $D_f$ for freshly emitted BC particles near the combustion source ranges from 1.6 to 1.9 (Gwaze et al., 2006). Transmission electron microscopy (TEM) analysis of  BC samples from different engines showed values for the fractal dimensions between 1.5 and 2.1 for diesel black carbon and 2.2 and 3.0 for spark-ignition engines (Wentzel et al., 2003). In this study, the value of $D_f$ in all the aggregate representations was set to 1.7, except for the sensitivity analysis. The $D_f$ of 1.7 is commonly representative of laboratory-generated fresh black carbon and was used after examining the TEM images from the mini-CAST generator provided in Ess et al. (2021).

A sensitivity analysis of various modelling parameters, like the refractive index, fractal dimension, and radius of the primary particle were conducted in this study to understand their relative importance towards the modelled optical properties. The results of the sensitivity study were focused on the bare particles from denuding experiment E1, excluding the impact of an organic coating. For studying the sensitivity of  $a_{pp}$, the optical properties were modelled for $a_{pp}$ ranging from 5 to 25 nm. In the sensitivity study of $D_f$, the optical properties are compared and validated for the 'aggregate' representation for $D_f$ ranging from 1.5 - 2.8. The dependency of the modelled optical properties on the real and imaginary parts of refractive index was also studied.  The optical properties were modelled using 'aggregate' and 'sphere' representations for the real part of the refractive index $m_{re}$ ranging from 1.2 to 2, and the imaginary part of the refractive index $m_{im}$ ranging from 0.2 to 1. For all the results of the sensitivity study, the modelled optical properties were compared with their laboratory equivalents for a better understanding of the subject.

### 2.3  Tools for modelling black carbon optical properties

Aggregation of black carbon agglomerates to form a larger fractal aggregate is described by the process of diffusion-limited cluster aggregation (Witten and Sander, 1983). Based on this principle, various Diffusion-limited algorithms (DLAs) have been developed. The tunable diffusion limited aggregation (DLA) software (Woźniak, 2012) was used in this study to simulate the 'aggregate', 'coated aggregate', and 'aggregate and sphere' BC representations. This algorithm preserves fractal characteristic of the aggregate, by iteratively adding each primary particle one by one.

The Multi-Sphere T-matrix Method (MSTM) code (Mackowski and Mishchenko, 2011) and the Lorentz-Mie theory (Hergert and Wriedt, 2012; Bohren and Huffman, 1998) were used to model the optical properties of simulated black carbon particles. The optical properties were calculated in the visible spectrum, for $\lambda$ equal to 467, 530, and 660 nm. It must be noted that the range of $\lambda$ was limited as only refractive index at the wavelengthss 467, 530, and 660 nm were available (Kim et al., 2015).

For the 'sphere', the 'homogeneously mixed sphere'and the 'coated sphere' representations, the Python Mie Scattering package PyMieScatt (Sumlin et al., 2018) based on the Lorentz-Mie theory was used. The MSTM code was used for the 'aggregate', 'coated aggregate', and 'aggregate and sphere' representations. The MSTM code contains a FORTRAN based algorithm that calculates the optical properties of a set of arbitrary spheres (Mishchenko et al., 2004). The MSTM code is therefore appropriate for computing the radiative properties of aggregates. The MSTM code has found wider applications in the research field because of better accuracy and comparatively lower computational cost for fractal like particle compared to other methods like the Discrete Dipole Approximation DDA (Liu et al., 2017).

The MSTM manual notes a limitation that the nested spheres in the particle should not intersect each other. However, in the case of 'aggregate and sphere' representation (Fig. 1f) the monomers of the aggregate intersected with the sphere at few points. The application of the MSTM code over particles with few intersecting spheres were tested by comparing them to the results of the Geometric Optics Surface-wave (GOS) approach used in the study by He et al. (2015). The results for the absorption cross section from both the methods were in good agreement with each other, summarized in the supplementary information of this manuscript. Therefore, the MSTM code was used for the case of 'aggregate and sphere' representation where few intersecting spheres were present.

The MSTM code and the Lorenz-Mie theory were used to calculate the extinction efficiency $Q_{ext}$, absorption efficiency $Q_{abs}$, scattering efficiency $Q_{sca}$, and the asymmetry parameter $g$. The asymmetry parameter $g$ is defined as the intensity-weighted average of the cosine of the scattering angle. The single scattering albedo (SSA) was further derived from the ratio of the scattering efficiency ($Q_{sca}$) to the extinction efficiency ($Q_{ext}$) as:

$$\text{SSA} = \frac{\sigma_{sca}}{\sigma_{ext}}. \tag{5}$$

The measured SSA was calculated using a combination of $\sigma_{scat}$ measured from nephelometer and extinction coefficient $\sigma_{ext}$ from the CAPS PM$_{ex\ 630}$. The mass absorption cross section of black carbon (MAC$_{BC}$) is calculated at a wavelength of 660 nm from the ratio of absorption cross section ($C_{abs}$) and BC mass (m$_{BC}$) as:

$$MAC_{BC} = \frac{C_{abs}}{m_{BC}} = \frac{C_{abs}}{\frac{1}{6}\pi d^3 \cdot \rho_{BC}}, \tag{6}$$

where $\rho_{BC}$ is the density of black carbon and taken in this study to be 1.8 g cm$^{-3}$ (Park et al., 2004).

The Ångström absorption exponent AAE describes the wavelength dependence of the aerosol light absorption. The AAE was calculated from the best fit of $\sigma_{abs}(\lambda)$ at the wavelengths $\lambda$ of 470, 520, and 660 nm by:

$$\sigma_{abs}(\lambda = 467, 530, 660\ nm) = C_o \lambda^{-AAE}, \tag{7}$$

where $C_o$ is a constant. It must be noted that the use of wavelengths $\lambda$ of 467, 530, and 660 nm for calculations is a result of the availability of the refractive indices nm (Kim et al., 2015) at which the modelled optical properties are calculated.

The absorption coefficient $\sigma_{abs}$ (unit: Mm$^{-1}$) is the sum of the absorption cross-section $C_{abs\_i}$ (unit: m$^2$) calculated for each available size range:

$$\sigma_{abs} = \sum_{d_i=1}^{d_n} C_{abs}(d_i) \cdot n(d_i), \tag{8}$$

where $n$ is the number concentration of the size range with diameter $d_i$. The absorption cross-section $C_{abs}$ is calculated from the absorption efficiency $Q_{abs}$ for each size range as:

$$C_{abs}(d_i) = Q_{abs}(d_i) \cdot \pi \frac{d_i^2}{4}, \tag{9}$$

Similarly, the scattering coefficient $\sigma_{sca}$ and the extinction coefficient $\sigma_{ext}$ are derived.

### 2.3.1 Size of the simulated black carbon particles

The optical properties were modelled for monodisperse and polydisperse number size distributions. The definitions for both the size distribution methods are given below:

- Monodisperse size distribution method: the optical properties were modelled for a single particle whose size was the mean diameter $d_{p,\bar{N}}$ of the number size distribution or the volume mean diameter $d_{p,\bar{V}}$ derived from the volume size distribution. The monodisperse size distribution method is commonly used in modelling studies of BC where the results are usually focused on single sized particles (e.g., Berry and Percival, 1986; Kahnert and Kanngießer, 2020; Smith and Grainger, 2014; Liu et al., 2018; Liu et al., 2019; Luo et al., 2018a).

-   Polydisperse size distribution method: the modelled optical properties are integrated over size according to the particle number size distribution. This ensemble-averaged size method is more relevant to ambient or laboratory studies of BC, where the optical properties are measured for a broad size distribution.

From this point forward, monodisperse and polydisperse size distribution methods will be referred to simply as 'monodisperse method and 'polydisperse method', respectively. Figure 2 provides an overview of Sec. 2, including the various experimental cases, morphological representations, and size distribution methods used to model the optical properties.

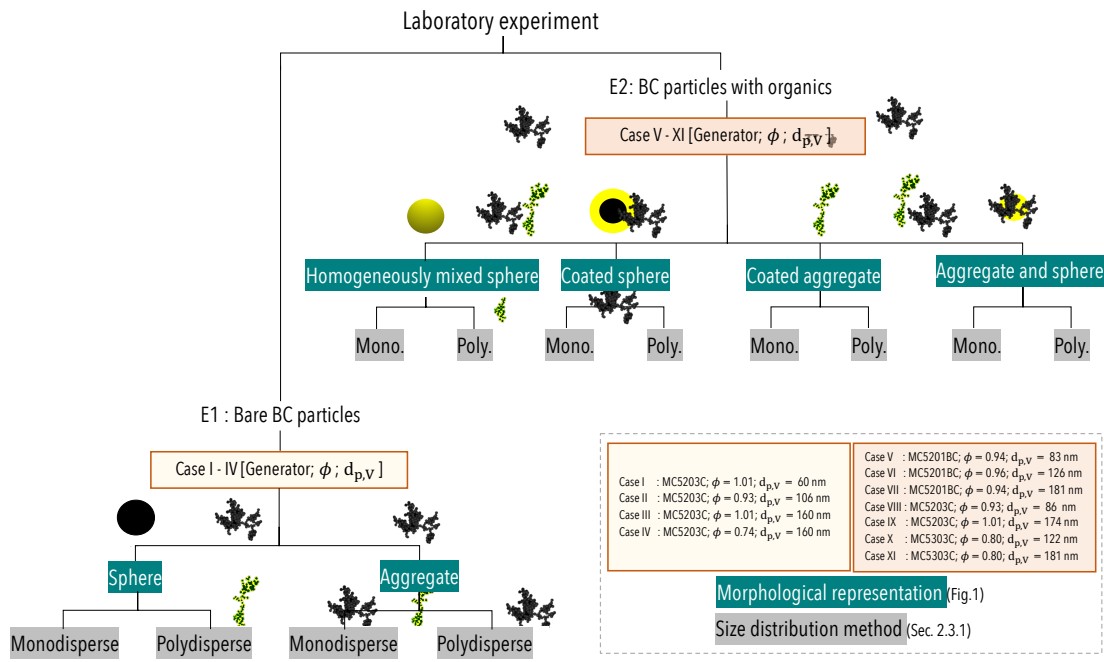


**Figure 2.** Schematic overview of the various experimental cases, morphological representations, and size distribution methods used to model the optical properties.

## 3  Results

### 3.1 Denuding experiment E1 - modelling techniques for bare BC

Figure 3 shows the single scattering albedo (SSA), and the Ångström absorption exponent (AAE) measured from denuding experiment $E_1$ at three heating conditions. All four cases showed that the SSA and AAE decreased as the particles were heated. It is expected that SSA and AAE will decrease as the volatile organic matter in the particles is removed during heating, leaving behind purer BC containing particles. However, in case IV($d_{p,\bar{v}} = 160$; $\phi = 0.74$), the heated particles showed relatively little change in their SSA, while almost negligible change in the AAE. It can be explained by the fact that the particles generated in case IV contain a lower amount of

volatile organic matter (Mamakos et al., 2013) due to the fact that it was a fuel-lean condition where < 1. Heating the particle under such fuel-lean conditions can result in relatively insignificant changes in the particle's SSA or AAE. Section 3.1 discusses the results of modelling techniques for pure BC. Because the particles are expected to have comparatively low organic carbon content, measurement results from the experiments with the catalytic stripper at 350 °C will be used for each case.

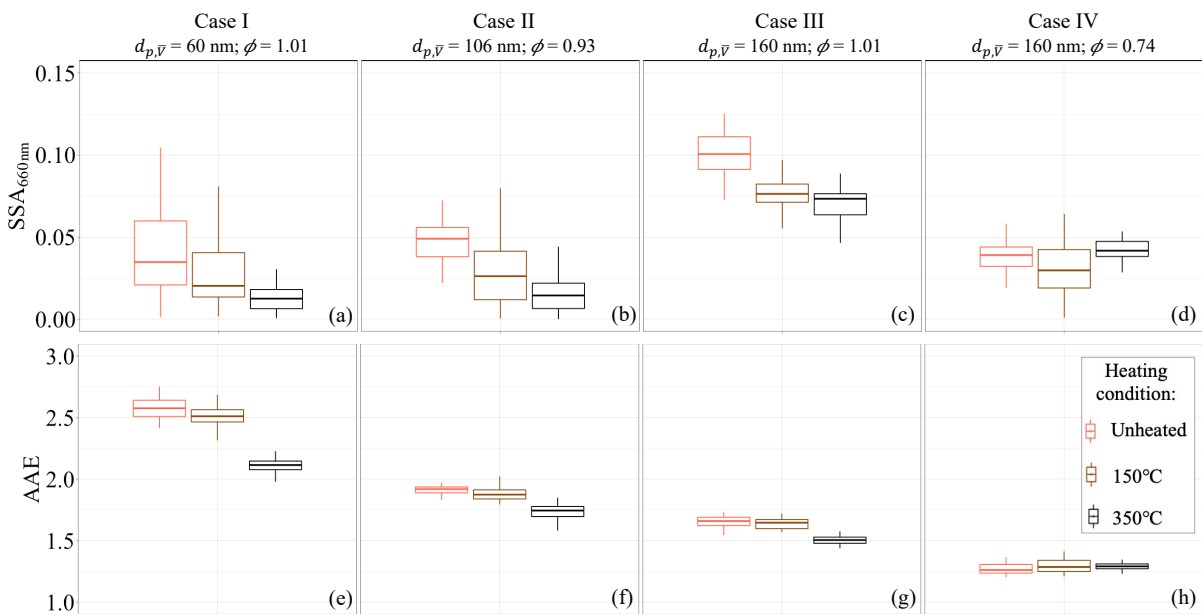


**Figure 3.** Measured optical properties for the four cases of experiment E1. Panels **(a-d)** show single scattering albedo (SSA), and **(e-h)** show Ångström absorption exponent (AAE). In each panel, the box-plots are arranged for three heating conditions where the BC particles bypass the Catalytic Stripper (unheated) or pass through the Catalytic Stripper operated at 150°C or 350°C, respectively. The operating condition is indicated in the legend on

the top-right of the figure.

### 3.1.1 Comparison of optical properties of monodisperse bare aggregates with different methods of calculating the primary particle number

Figure 4 compares the SSA modelled using the three different methods available for estimating the number of primary particles ($N_{pp}$) in an aggregate. For each case of E1, the three methods were compared using both diameters $d_{p,\bar{N}}$ and $d_{p,\bar{V}}$. The modelled SSA from the three methods is compared with the experimentally measured SSA for each case. For the results of $d_{p,\bar{N}}$, the modelled SSA calculated using the three methods showed a variability of up to a factor of two with respect to each other. In comparison, the difference in the SSA increased

to a factor of 2.8 for the results of $d_{p,\bar{V}}$. Based on the comparison of the modelling results with the measured SSA, it was evident that the performance of each method differs depending on particle characteristics (i.e., different for each case of E1). As a result, one single method could not be recommended. The method by Sorensen (2011), however, was used as the standard method since it involved the fewest assumptions.

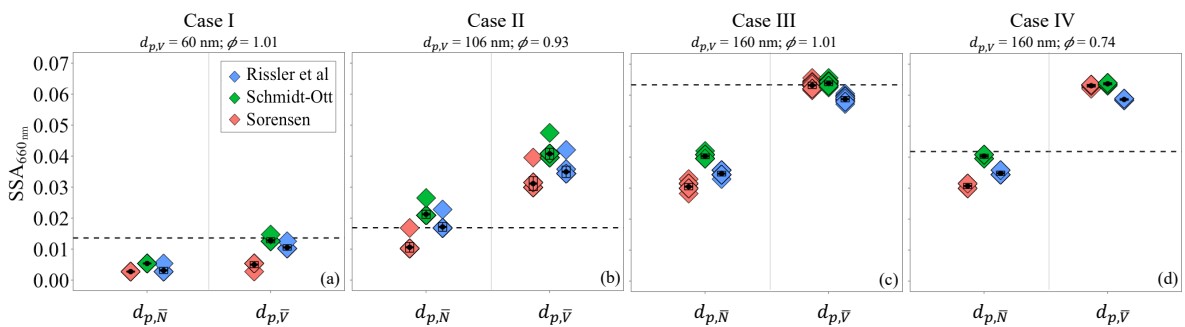


**Figure 4.** Modelled single scattering albedo (SSA) of bare BC aggregates using the three methods for calculation of the primary particle number (Rissler et al., 2012; Sorensen, 2011; and Schmidt-Ott, 1988). Panels **(a-d)** show the results for the four cases of E1. For each case, the three methods were applied to calculate the $N_{pp}$ using both

the $d_{p,\bar{N}}$ and $d_{p,\bar{V}}$ (x-axis). The mean of the modelled *SSA* for each method is shown by the black point. The dashed line in the panels represents the mean of the experimentally measured *SSA*.

### 3.1.2 Optical properties of spherical and fractal bare BC particles using the monodisperse method

Figure 5 shows the modelled optical properties for the four cases in experiment E1 using a monodisperse method. In each case, the optical properties of the laboratory-generated particles were modelled assuming both a spherical and an aggregate representation. The modelled SSA (Fig. 5a-d) was mostly in the range of the experimentally measured SSA when using an aggregate representation. When the particles $\lesssim$ 100 nm, as in case I or case II, the modelled SSA using the spherical representation also was found in the range of the measured values. However, when the particles are larger than 100 nm using spherical assumption overestimated the modelled SSA by up to a factor of 2 to 5. When using a spherical representation, the SSA may be overestimated due to the short residence time of the particle generated in the lab, where these particles are unlikely to be spherical or compact. Previous studies have also noted an increase in scattering as the particles becomes more compact in shape (Luo et al., 2018b; Yuan et al., 2020; Li et al., 2016).
Ångström absorption exponent

**Figure 5.** Optical properties of bare BC particles modelled using the monodisperse method compared to their measured values. Panels **(a-d)** show single scattering albedo (SSA), and **(e-h)** show Ångström absorption exponent (AAE) for the cases I – IV of E1. In each panel/case, the 'sphere' and 'aggregate' representation for bare BC particles was used as shown in the legend. The SSA and AAE is modelled using both $d_{p,\bar{N}}$ and $d_{p,\bar{V}}$ (X-axis). The shaded yellow area represents the experimentally measured values, with the dashed line being the mean of the measured SSA or AAE. The lower hinge and the upper hinge of the boxplot represent the 25% and 75% quantile of the observations, respectively. The lower whisker is equal to the smallest observation greater than or equal to lower hinge - 1.5*IQR. Similarly, the upper whisker is equal to the largest observation less than or equal to upper hinge + 1.5*IQR. The meaning of these terms is consistent for boxplots through this study.

The absorption coefficient ($\sigma_{abs}$) was modelled more accurately when using an aggregate representation and the mean diameter of the number size distribution $d_{p,\bar{N}}$ (Fig. 5e-h). Similar to the modelled SSA, for cases with smaller particles, there was a minimum difference between the aggregate and spherical representations in the

modelled $\sigma_{abs}$. In the case when the $\sigma_{abs}$ is modelled using a monodisperse size distribution of particles with $d_{p,\overline{V}}$, the results are overestimated by up to a factor of four. It is apparent from this result that assuming the size distribution to be monodisperse may lead to an overestimation of the total light absorption. The assumption that the monodisperse population has a mean diameter of the number size distribution, however, is suggested for better results in the case of a modelled $\sigma_{abs}$. The mass absorption cross-section (MAC$_{BC}$) was calculated for the four cases of the experiment E1. Modelled values of MAC$_{BC}$ ranged from 2.44 to 4.66 m$^2$/g when using pure BC. Because of the unavailability of an instrument directly measuring the mass in E1, the mass was calculated assuming a density of 1.8 g cm$^{-3}$ (Park et al., 2004). In smaller BC particles ($d_{p,\overline{V}}$ of 60 and 106 nm), the modelled MAC$_{BC}$ is larger than the measured MAC$_{BC}$. This may be because of the reason that the smaller particles contain higher residual organic matter (Zhang et al., 2020), which results in an underestimation of the measured MAC$_{BC}$ when a density of 1.8 g cm$^{-3}$ is used. The results for modelled MAC$_{BC}$ is provided in the supplementary material of this manuscript.

The modelled AAE (Fig. 5i-l) was underestimated for particles $\lesssim$ 100 nm in both cases I and II, for both spherical and aggregate representations. The results of Zhang et al., 2020, indicate that smaller particles contain a greater amount of organic carbon than larger particles, which makes the removal of all organic carbon with Catalytic Stripper more challenging. There may be residual nonvolatile organic matter in smaller particles, which results in a lower modelled AAE when such particles are assumed to be purely BC. The AAE could not be accurately modelled without information regarding the chemical composition of the BC particles, even if they have been denuded. Liu et al. (2018) found that the modelled AAE for bare BC particles was higher in the case of aggregate morphology than when a spherical structure was assumed. However, the results of the modelled AAE in this study showed a size dependency. It can be observed that the AAE values increase as the particle size increases from case I to case II. The AAE values, however, decrease when the particle size increases further in case IV. Similarly, the comparability of aggregate and spherical results was influenced by the particle size. An explanation of the size-dependence of the AAE is provided in Fig. 12 of Romshoo et al., 2021. It was difficult to suggest a single method of modelling AAE due to size-dependence and residual organics, however the spherical assumption was in better agreement in some cases.
modelling

### 3.1.3 Optical properties of spherical and fractal bare BC particles using the polydisperse method

Figure 6 shows the optical properties of the four cases in experiment E1 using a polydisperse method. Based on the measured polydisperse number size distribution, the optical properties were modelled using spherical and aggregate representations. For all the four cases of E1, the modelled SSA (Fig. 6a-d) matched more closely to the experimentally measured values when using the aggregate representation. In the polydisperse method, the modelled SSA was overestimated by nearly three times when using a spherical representation for particles larger than 100 nm. The explanation for the overestimation when using spherical assumption was explained in the section 3.1.2.

The modelled $\sigma_{abs}$ is compared to the experimentally measured $\sigma_{abs}$ from the AE33 instrument in Fig. 6e-h. It was observed that the modelled $\sigma_{abs}$ using the aggregate representation was in good agreement with the measurements for all the four cases of the experiment. Comparatively, using the spherical assumption overestimated the modelled $\sigma_{abs}$ by nearly 1.6 to 3, as a function of particle size. He at al. (2014) showed that the absorption modelled for monodisperse BC particles using an aggregate representation is still 25% less, when compared to measured values. In this study it was demonstrated that this discrepancy between modelled and measured absorption results can be reduced to 10%, when using the polydisperse method, in combination with an aggregate representation of black carbon.

Figure 6(i-l) compares the modelled AAE with the experimentally measured values. As discussed in the previous section, the results from the modelled AAE differ from the measurements as a function of size. With the polydisperse method, both spherical and aggregate representations underestimate the AAE in case I of smaller particles containing a higher level of residual organic matter. Contrary to this, in case IV, when there is anticipated to be less organic matter present, both aggregate and spherical representations model the AAE within the measurement range. For all the optical properties, in general, the discrepancies when using the monodisperse method was comparatively larger to when the polydisperse method was used. The optical properties were also modelled at other wavelengths in the visible range and compared to their respective measured values. The modelled values for other wavelengths also followed similar trends as those shown in Figure 6. The results are shown in the supplementary material to this manuscript.

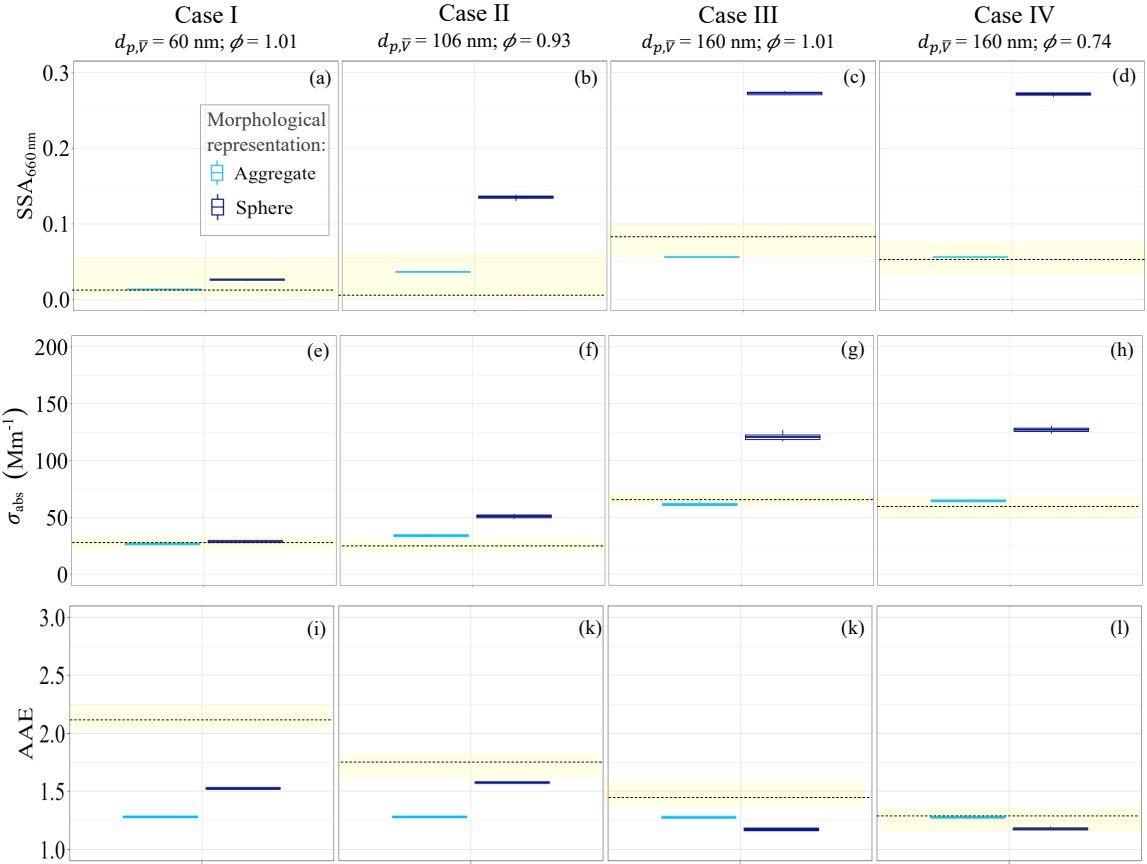

**Figure 6.** Optical properties of bare BC particles modelled using the polydisperse method compared to their measured values. Panels **(a-d)** show single scattering albedo (SSA), and **(e-h)** show absorption coefficient ($\sigma_{abs}$) for the cases I – IV of E1. In each panel/case, the 'sphere' and 'aggregate' representation for bare BC particles was used as shown in the legend. The shaded yellow area represents the experimentally measured values, with the dashed line being the mean of the measured SSA or $\sigma_{abs}$.

### 3.1 Modelling techniques for bare BC – sensitivity study

### 3.2.1 Refractive index

Figure 7 compares how the optical properties of bare BC particles vary with complex refractive index for two different morphologies. This sensitivity study was performed on case III ($d_{p,\bar{V}} = 160$ nm), where Figures 7a and 7c show the modelled SSA and AAE when the real part of the complex refractive index ($m_{re}$) was varied between 1.2 to 2. Particle light absorption is generally associated with the imaginary part of the RI However, our results showed that the absorption also depends on the real part of the RI, especially for spherical particles (Fig. 7c). SSA and AAE were both shown to be more sensitive to the real part of the RI for spherical morphology. As a result, with changes in the real part of the RI, the SSA and AAE of spherical morphology differed by factors of three and two, respectively. The SSA and AAE calculated using the aggregate morphology were less sensitive to changes in the real part of RI, and showed good agreement with the measured results when ranging between 1.6 and 2. Previously, Liu et al. (2018) also reported that the AAE of fresh BC aggregates depends very little on the complex refractive index, but when the aggregate particles are compact or coated, their sensitivity to the imaginary part of the RI increases.

Figures 7b and 7d show the dependence between optical properties of bare BC particles and the imaginary part of the RI ($m_{im}$) for different morphologies. For the imaginary part of the RI as well, the SSA and AAE spherical calculated with the spherical morphology were more sensitive. AAE varied by a factor of three with changes in the imaginary part of the RI, as is expected. There was an interesting observation that the SSA decreased as the imaginary part of the RI was increased up to 0.5, after which there was an increase in the SSA (Fig. 7b). This behavior was observed only in the case of spherical particles. In comparison, SSA calculated with an aggregate representation decreased with the imaginary part of the RI, and it was in good agreement with the measured results

when the imaginary part of the RI was between 0.3 and 1. He et al., 2015 also showed that the optical properties of BC aggregates can vary up to a factor of 1.6 due to changes in the refractive indices, which is in agreement with the results in this study. It is interesting to see that the dependency of the particle light absorption over the complex refractive index varies depending on whether spherical or aggregate representations are used.


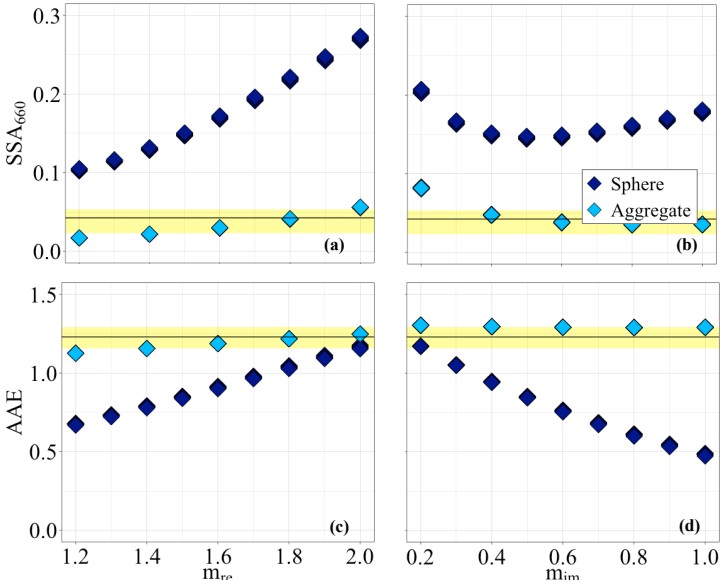

**Figure 7.** Optical properties of bare BC particles studied as a function of real part and imaginary part of the complex refractive index ($m_{re}$ and $m_{im}$) . The results are shown for *SSA* **(a, b)** and AAE **(c, d)** for the case *III* ($d_{p,VED}$ = 160 nm) of E1 using both 'sphere' and 'aggregate' representation. The yellow area in the figure
represents the experimentally measured values, and the dashed line is the mean measured value for each case. Panels **(a, c)** are defined by fixed imaginary parts, while panels **(b, d)** are defined by fixed real parts according to Table B1.


### 3.2.2 Fractal dimension

Figure 8 shows the dependency of the optical properties towards fractal dimension ($D_f$). As discussed in section 2.1, black carbon fractal aggregate can have a wide range of $D_f$ depending on the source of combustion, chemistry during formation, and the atmospheric conditions. To determine how sensitive the SSA and AAE are, the fractal dimension was varied between 1.5 and 2.8 for four cases of experiment E1. In both of the first two cases, when the particle size does not exceed 100 nm, the modelled SSA is the least sensitive to changes in the $D_f$. When the
BC particle is small and rather fresh, the change in the $D_f$ has relatively less significance.
The results of previous studies also indicate that in particles with a mobility diameter less than 100 nm, the effect of fractal dimension on SSA is negligible (Fig. 6e from Romshoo et al., 2021). When particles were larger than 100 nm, the modelled SSA. showed variability of up to 100% for the polydisperse method. Further, the sensitivity of the AAE (Fig 8e-h) towards the fractal dimension was very less, especially in smaller particles, for both
monodisperse and polydisperse size method. The dependence of the AAE over the $D_f$ as a function of particle size can be seen in detail in Figure 12 of Romshoo et al., 2021. For this study, it was found that the modelled SSA agreed well with the measured values when the fractal dimension ranges between 1.7 and 1.9, which are characteristic values for non-aged BC (Gwaze et al., 2006) and may be applicable to mini-CAST generators.


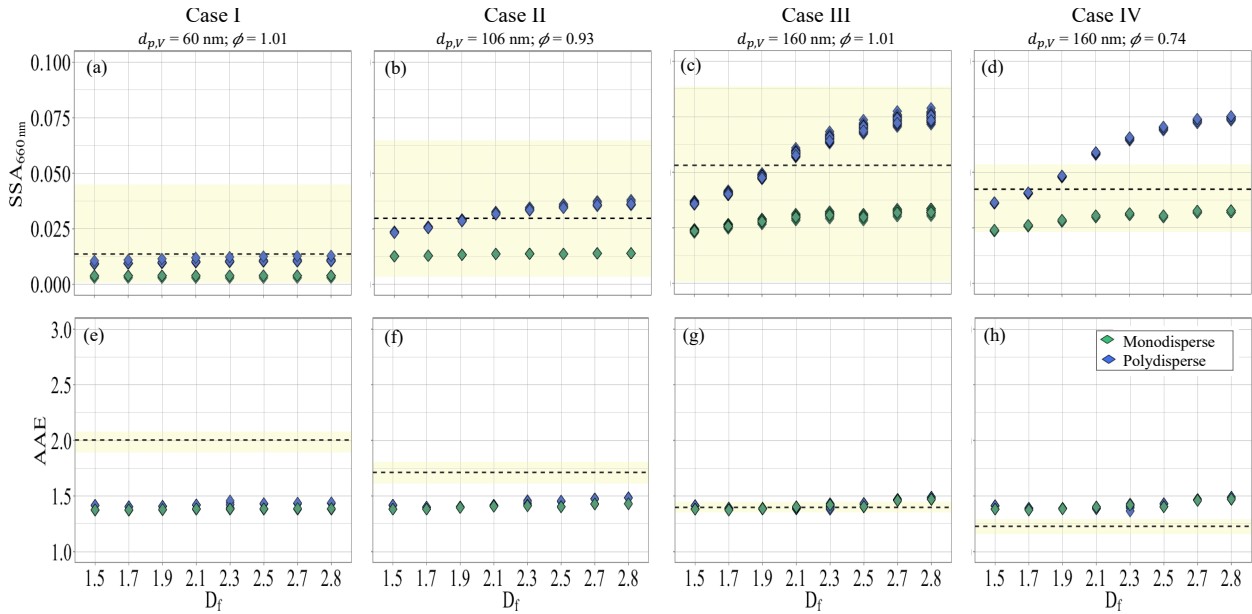

**Figure 8.** Sensitivity analysis of the modelled SSA and AEE using the 'aggregate' representation. The fractal dimension $D_f$ was varied between 1.5 to 2.8 for cases I – IV of E1, modelling the SSA **(a to d)** and AEE **(e to f).** The experimentally measured values are highlighted by the yellow area in the figure, and the dashed line represents the mean measured SSA or AEE for each case.

### 3.2.3   Primary particle radius

Figure 9 shows the how sensitive the optical properties are towards the primary particle size ($a_{pp}$) for monodisperse and polydisperse size method. For each case of experiment E1, the $a_{pp}$ was varied between the range of 5 to 28. Earlier studies have reported that the optical cross-sections are not sensitive towards the primary particle size (He et al., 2014). However, the particle light scattering showed a dependence on the $a_{pp}$, as the SSA varied by a factor of three as a function of the $a_{pp}$ (Fig. 9 b-d). When the SSA is modelled using the polydisperse approach, the dependence is more pronounced and increases to a factor of six. It was shown that when the $a_{pp}$ is between 10 and 14 nm, the modelled SSA is in good agreement with the measured SSA for all cases. It is therefore recommended that for future studies $a_{pp}$ between the range of 10 and 15 nm be used . When compared the dependency of modelled AAE towards $D_f$ (Fig. 8e-h), the modelled AAE was observed to be more sensitive to $a_{pp}$ (Fig. 9e-h). In the case of AAE, the monodisperse and the polydisperse method showed similar dependency towards $a_{pp}$. With the exception of case III, where larger particles and low residual organics are present, the optical model was not able to reproduce the measured results of the AAE. A discussion of possible reasons for discrepancies in modelled AAE was provided in Section 3.1.2.

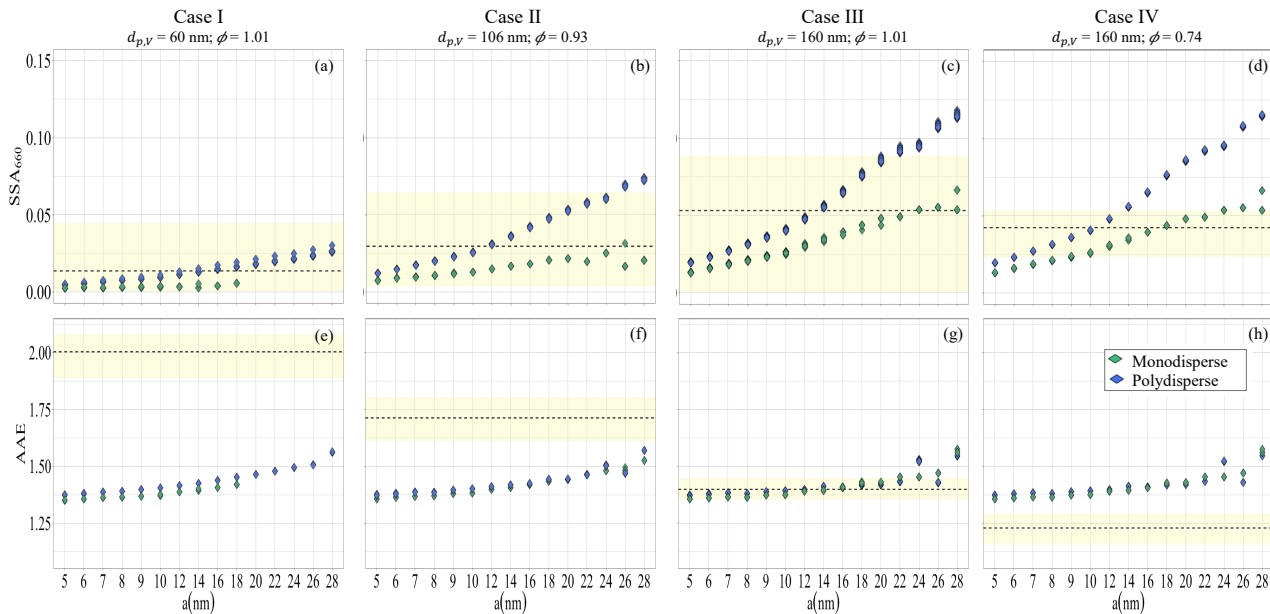

**Figure 9.** Sensitivity analysis of the modelled SSA and AEE using the 'aggregate' representation. The primary particle radius $a_{pp}$ was varied between 5 to 28 nm for cases I – IV of E1, modelling the SSA **(a to d)** and AEE **(e to f).** The experimentally measured values are highlighted by the yellow area in the figure, and the dashed line represents the mean measured SSA or AAE for each case. There are missing points in **(a)** and **(e)** of the monodisperse method results, which indicates that the particles are too small to form an aggregate with $a_{pp} > 18$ nm.

### 3.3 Experiment E2 - modelling techniques for BC with organics

This section discusses the modelled optical properties of BC particles containing organics (experiment E2), comparing various morphological representations and size methods used for modelling. .Figure 10 shows the results of the modelled SSA for the different cases of the mini-CAST generators used in experiment E2 (Table 1). For each case of E2, the SSA was modelled using four representations of BC particles with organics: 'coated sphere', 'homogeneously mixed sphere', 'coated aggregate', and 'aggregate and sphere'. Further, the SSA is modelled for both polydisperse and monodisperse methods.

In the case of mini-CAST 5201BC and 5303C (Fig. 10a, 10d, 10c, and 10f), the SSA modelled using both the aggregate representations('coated aggregate', 'aggregate and sphere') agreed well with the measurements for all the size methods. However, for one of the cases of mini-CAST 5203C (Fig. 12e), the results of the aggregate representations underestimated the SSA. It was noted that the sensitivity of the modelled SSA to the various morphological representations become comparatively less prominent in the case of smaller particle size (Fig. 10b, $d_{p,\bar{v}} = 86$ nm, for mini-CAST 5203C). This case is similar to the outcome of smaller pure BC particles in experiment E1, where the modelled SSA did not depend on the fractal dimension (Fig. 8a).

In all the cases of E2, using the monodisperse method (with $d_{p,\bar{v}}$) and polydisperse method for spherical representations ( 'coated sphere', 'homogeneously mixed sphere') overestimated the SSA by up to a factor of three. Overall, the SSA modelled using the 'coated aggregate' representation matched the measured values most closely, with a maximum deviation of 20% in certain cases. In the theoretical study by Liu et al. (2018), where absorption cross-section of coated BC was modelled, it was shown that the dependence on the morphology was size-dependent and wavelength-dependent. There was a similar size-dependence between the morphology and the modelled SSA, for e.g., for the two cases of mini-CAST 5203C (Fig. 10b, 10e). When the particle is smaller in size (86 nm), the SSA calculated for the 'sphere and aggregate' representation using the monodisperse method is higher than that for the polydisperse method (Fig. 10b). In contrast, when the particle is larger (174 nm), the SSA calculated from the polydisperse method is larger than the one calculated using the monodisperse method (Fig. 10b).

Interestingly, the SSA calculated using the 'aggregate and sphere' representation showed comparable results to that calculated using the 'coated aggregate' representation. Although laboratory-generated black carbon is less likely to resemble the 'aggregate and sphere' depiction since the organic mass is evenly distributed throughout the BC aggregate. This representation usually depicts an aged black carbon particle where the BC aggregate is entirely encapsulated in a sphere of coating. Therefore, it is expected that using it for representing laboratory-generated black carbon would create a lensing effect, simulating higher absorption. However, because the coating

accounted for less than 55% of the total particle volume, in none of our cases the coating encapsulated the
aggregate. When the volume of coating is larger in laboratory-generated black carbon, using the 'aggregate and
sphere' representation may overestimate the absorption because of the lensing effect.

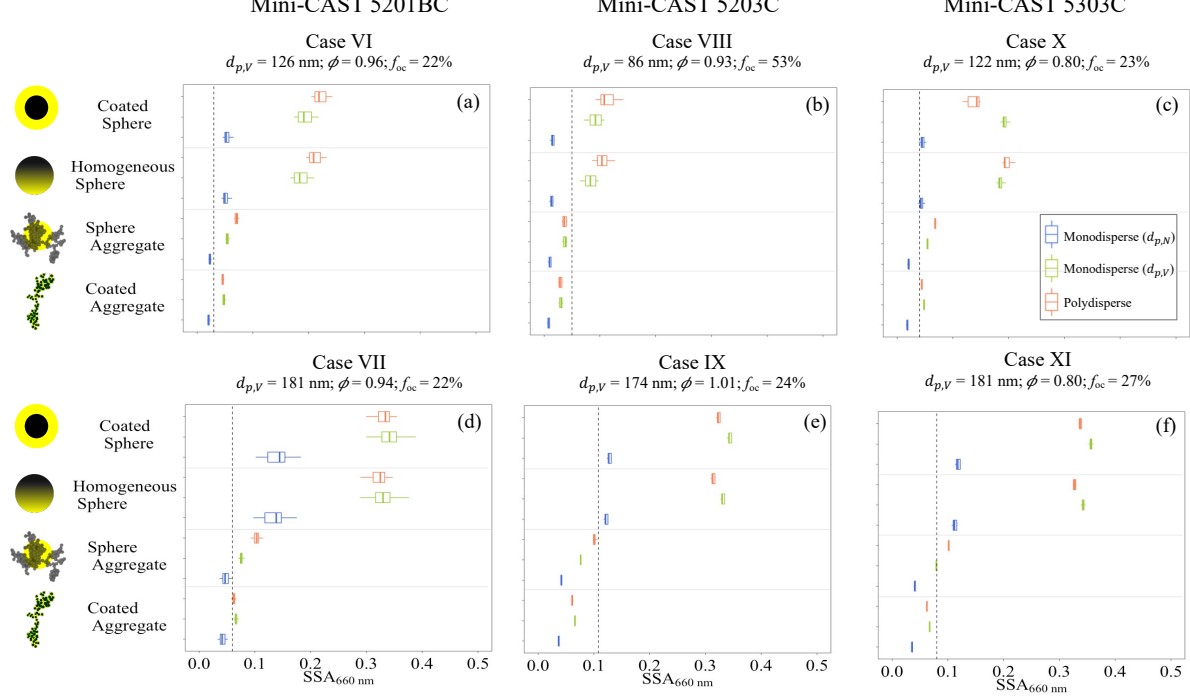

**Figure 10.** Modelled SSA at a wavelength of 660 nm from various cases of mini-CAST generators in E2
summarized in Table 1. The results are shown for: mini-CAST 5201BC $d_{p,\bar{V}}$ = 126 nm **(a)**; mini-CAST 5201BC
$d_{p,\bar{V}}$ = 181nm **(d)**; mini-CAST 5203C $d_{p,\bar{V}}$ = 86 nm **(b)**; mini-CAST 5203C $d_{p,\bar{V}}$ = 174 nm **(e)**; mini-CAST 5303C
$d_{p,\bar{V}}$ = 122 nm **(c)**; mini-CAST 5303C $d_{p,\bar{V}}$ = 181 nm **(f)**. In each panel, the SSA is modelled using four coated
BC representations 'coated sphere', 'homogeneously mixed sphere', 'coated aggregate', and 'aggregate and
sphere'. Further, for each representation the SSA is modelled using monodisperse and polydisperse method. The
mean of the experimentally measured SSA is shown by the black dashed line in each panel.

Figure 11 shows the modelled asymmetry parameter *g* for three cases of mini-CAST 5201BC. For each case,
the *g* was modelled using the four representations of coated BC i.e., 'coated sphere', 'homogeneously mixed
sphere', 'coated aggregate', and 'aggregate and sphere'. Further in each of the representation, the *g* was calculated
for monodisperse method. It was observed that the value of *g* increased as the coated BC particle grow in size,
indicating more forward scattering for larger BC particles  (Fig. 11a to 11c). However, the rate of increase of
forward scattering with growing BC particles was more evident in the aggregate representations. Due to lack of
experimental measurement of *g*, the modelled results could not be validated with the measured findings. There
have been previous studies estimating asymmetry parameter *g* from nephelometers, however, this calculation is
very uncertain. For example, the simple parameterizations using the hemispheric backscatter fraction (Andrews
et al., 2006) were derived for ambient and more spherical aerosol particles. It is not clear how this parameterization
works for BC aerosols with low single scattering and fractal morphology. The limitations of the Aurora 4000
nephelometer (Müller et al., 2012) used in this study is that the polar function is measured in up to 18 angular
sectors in forward scattering direction, whereas the real resolution is smaller, since the shutter function is not steep
enough. Furthermore the hemispheric backscattering is just represented as one large sector (scattering angle 90 to
180°). There is still a need to examine Aurora4000 in more depth to determine an asymmetry parameter for fractal
soot particles.

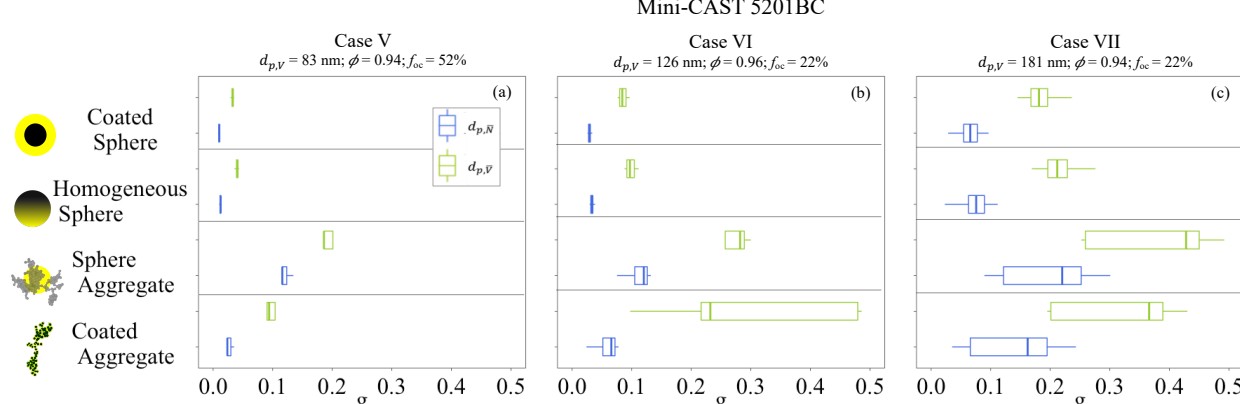

**Figure 11.** The asymmetry parameter $g$ is modelled using the four representations of coated BC i.e., 'coated sphere', 'homogeneously mixed sphere', 'coated aggregate', and 'aggregate and sphere'. For each representation, the $g$ is modelled using monodisperse particles (with $d_{p,\bar{N}}$ and $d_{p,\bar{V}}$). The results are shown for E2 cases V – VII: mini-CAST 5201BC $d_{p,\bar{V}}$ = 83 nm **(a)**; mini-CAST 5201BC $d_{p,\bar{V}}$ = 126 nm **(b)**; and mini-CAST 5201BC $d_{p,\bar{V}}$ = 181 nm **(c)**.

Figure 12 shows the BC mass absorption cross-sections (MAC$_{BC}$) modelled for three different cases of mini-CAST 5201BC ($d_{p,\bar{V}}$ = 83, 126, and 181 nm). In each case, MAC$_{BC}$ is modelled using the four representations of coated BC i.e., 'coated sphere', 'homogeneously mixed sphere', 'coated aggregate', and 'aggregate and sphere'. Forestieri et al. (2018) found that the spherical assumption used in the Lorentz-Mie theory underestimates the modelled mass absorption cross-sections (MAC$_{BC}$) for bare flame-generated black carbon. Figure 12a ($d_{p,\bar{V}}$ = 83 nm, and $f_{oc}$ = 64%) shows that the MAC$_{BC}$ calculated using spherical and aggregate representations underestimated the MAC$_{BC}$, consistent with Forestieri et al. (2018). However, for larger $d_{p,\bar{V}}$, the spherical representations overestimated the MAC$_{BC}$ (Fig. 14b and 14c). In general, for larger particles, the modelled MAC$_{BC}$ and measured MAC$_{BC}$ were in better agreement when using 'coated aggregate' representation.

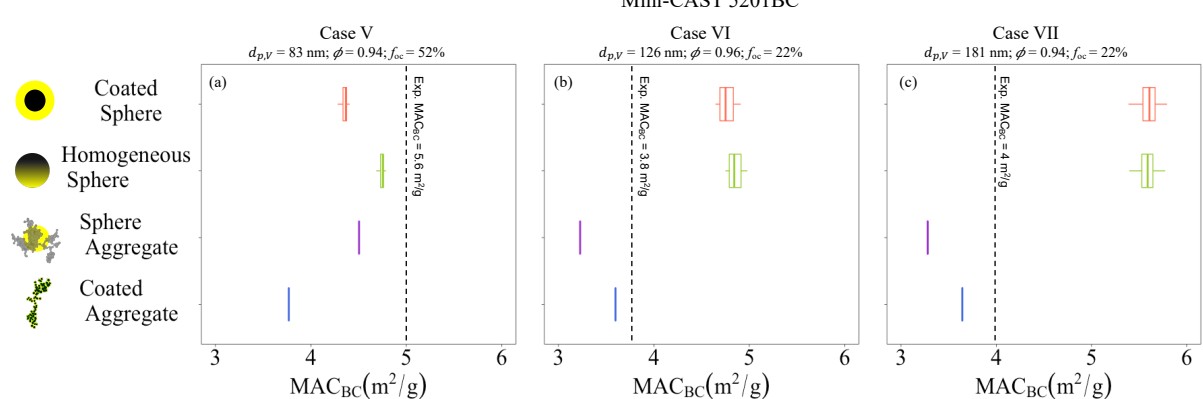

**Figure 12.** The BC mass absorption cross-section (MAC$_{BC}$) is modelled using the four representations of coated BC i.e., 'coated sphere', 'homogeneously mixed sphere', 'coated aggregate', and 'aggregate and sphere'. The results are shown E2 cases V – VII: mini-CAST 5201BC $d_{p,\bar{V}}$ = 83 nm **(a)**; mini-CAST 5201BC $d_{p,\bar{V}}$ = 126 nm **(b)**; and mini-CAST 5203C $d_{p,\bar{V}}$ = 181 nm **(c)**.

## 4 Conclusions

This work investigates the various modelling techniques for the optical properties of black carbon; and based on the results, recommendations for representing the morphology and size of black carbon are provided to the scientific community. The main goal of this study is to validate the different modelling approaches; therefore, the modelled optical properties were compared to measurements from laboratory-generated black carbon. The study is divided into two parts: (1) modelling techniques for bare BC – experiment E1; and (2) modelling techniques for BC with organics – experiment E2.

The laboratory experiment E1 was designed in such a way as to provide us with data to study modelling approaches for bare BC particles. The aerosol generated with a Catalytic Stripper is operated at 350°C is expected

to have the lowest organic content. Therefore, this condition was considered most suitable for modelling the optical properties of bare BC. For modelling the optical properties of bare BC, the two morphological representations 'sphere' and 'aggregate' were compared. Further for each morphological representation of bare BC, the optical properties were modelled using two size methods: for monodisperse particles (monodisperse method), and for polydisperse particles (polydisperse method).

For both the size methods, the modelled SSA was mostly in the range of the experimentally measured SSA when using an aggregate representation. When the particles $\lesssim$ 100 nm, the modelled SSA using the spherical representation also was found in the range of the measured values. However, using a spherical assumption overestimated the modelled SSA by a factor of two to five when the particles are larger than 100 nm. Using a spherical assumption may result in an overestimation of the SSA since the particles are unlikely to be spherical or compact due to their short residence time in the lab. It has also been noted in previous studies that scattering increases as shape of particles becomes more compact (Luo et al., 2018b; Yuan et al., 2020; Li et al., 2016). It was observed that the modelled $\sigma_{abs}$ calculated using the aggregate representation was also in good agreement with the measurements for all four cases of the experiment. On the contrary, using the spherical assumption overestimated the modelled $\sigma_{abs}$ by nearly 1.6 to 3, as a function of particle size. Overall, in the case of polydisperse particles, the modelled $\sigma_{abs}$ and SSA using the aggregate representation were in excellent agreement with the measured optical properties. Moreover, it was shown that the discrepancies between modelled and laboratory measured absorption can be reduced to 10%, when the combination of polydisperse method and an aggregate representation of BC is used. These results indicate that the mini-CAST generated black carbon particles are indeed fractal-like as also shown by TEM images (Ess et al., 2021; Mamakos et al., 2013) and aggregate representation of BC is recommended to be used when modelling their $\sigma_{abs}$ and SSA.

However, it was difficult to suggest a single method for modelling AAE due to size-dependence and residual organics, although the spherical assumption was in better agreement in some cases. It was observed that the AAE values increased as the particle size increased till 100 nm. The AAE values, however, decreased as the particle size increased beyond 100 nm. The particle size also impacted the comparability of aggregates and spheres. In the case that the smaller particles were immature solid black carbon with embedded organic content, the assumption that they are bare may account for the underestimation of modelled AAE in comparison to measured values. For further studies it would be useful to have EC/TC measurements in such an experiment in order to determine the absolute uncertainty in terms of the residual organic matter when the Stripper is used with optical instruments.

After studying the various size and morphological representations for modelling bare BC particles, the assumptions of various modelling parameters (for e.g., $m_{re}$, $m_{im}$, $D_f$, and $a_{pp}$) were evaluated. The sensitivity of the modelled optical properties (SSA and AAE) to the real and the imaginary part of the complex refractive index ($m_{re}$ and $m_{im}$) was studied. Light absorption by particles is commonly associated with the imaginary part of the RI, but in our study, we found that the absorption also depends on the real part of the RI. Both SSA and AAE showed a greater sensitivity to the real part of the RI for spherical morphology. Consequently, with changes in the real part of the RI, the SSA and AAE of spherical morphology differed by a factor of three and two, respectively. Depending on whether spherical representations or aggregate representations are used, we can observe different relationships between particle light absorption and complex refractive index. The modelled optical properties of BC were well aligned with measured values when using the aggregate morphological representation and assumptions of refractive indices as: (i) $m_{re}$ between 1.6 to 2; and (ii) $m_{imag}$ between 0.50 to 1.

In certain cases, using aggregate morphological representations of black carbon results in more accurate optical properties, so we investigated their sensitivity to two key aggregate parameters, the fractal dimension $D_f$ and primary particle radius $a_{pp}$. It was found that the modelled SSA was least sensitive to changes in fractal dimension when particle size was below 100 nm. The changes in fractal dimension were also less dependent on particle scattering calculated using the monodisperse method. As a consequence, when the BC particle is small and rather fresh, the change in the fractal dimension is of relatively less significance. In contrast, SSA modelled using the polydisperse method showed variability of up to 100% for particles larger than 100 nm. For both monodisperse and polydisperse size methods, the AAE showed a low sensitivity to the fractal dimension, particularly for smaller particles. Previous studies have indicated that optical cross-sections are not affected by the size of the primary particle (He et al., 2014). There was, however, a significant relationship between particle light scattering and particle size, with the SSA increasing by a factor of three as the particle size increased. In the polydisperse method, the dependence of the SSA is more apparent and increases by a factor of six. It was found that the modelled and experimentally measured optical properties of BC agree well when: (i) $D_f$ from 1.7 to 1.9, and (ii) $a_{pp}$ between 10 to 14 nm.

In order to study the modelling approaches for BC particles containing organics, three kinds of mini-CAST black carbon generators were used to produce black carbon particles with organic carbon content between 35 - 65%. Four kinds of morphological representations for coated BC (two each for spherical and aggregate) were compared using both monodisperse and polydisperse particles. In the most of the results, the modelled SSA using the 'coated aggregate' and 'aggregate and sphere' representation was in good agreement with the measured SSA. Though it is less likely that laboratory-generated black carbon will resemble the 'aggregate and sphere'

representation, it can still be used when the coating only makes up a small part of the total particle volume. Therefore, our results show that for coated black carbon particles as well, the aggregate morphological representation gives more accurate modelled SSA. Furthermore, when the polydisperse method is used, accuracy can be increased by up to two times. Similar to results for pure black carbon, the modelled AAE showed larger discrepancies, but matched the measured house in some instances when it was modelled using a spherical assumption. For $MAC_{BC}$ as well, both spherical and aggregate representation underestimated the $MAC_{BC}$ for smaller particles, though, the homogenous sphere representation was comparatively closer to the measured $MAC_{BC}$. For particles larger than 100 nm, the $MAC_{BC}$ was modelled more accurately when using the aggregate representation.

In general, the aggregate representation performed well for modelling the $\sigma_{abs}$, SSA, and $MAC_{BC}$ for laboratory generated BC particles with $f_{oc}$ less than 53% and $d_{p,V}$ larger than 100 nm. Whereas, the spherical representation performed well for modelling the AAE in larger particles ($d_{p,V} > 100$ nm). However, for smaller particles, using both aggregate or spherical representation results in a larger discrepancy when modelling the AAE or $MAC_{BC}$. The discrepancy was more pronounced in the cases of experiments E1, where the EC/TC analysis was not conducted. Therefore, the discrepancy could be a result of the presence of organic matter in smaller particles even after being heated by the Catalytic Stripper. The presence of larger percentage of organic matter in smaller particles is also observed from the results of the EC/TC analysis of experiment E2 where the largest $f_{oc}$ was observed in case with the smallest $d_{p,V}$ of 86 nm ($d_{p,N} = 48$ nm). The smaller particles were also found to be less sensitive to input parameters such as fractal dimension and primary particle size, making identification of the source of the uncertainty more difficult. Together, these results emphasize the importance of morphology and size representation in accurately modelling the optical properties of BC particles, and the need for further investigation to achieve an accurate model.

This study provides experimental support for previous theoretical work based on BC as fractal aggregates (e.g., Kahnert, 2010; Adaichi et al., 2010; Kahnert and Kanngießer, 2020; Smith and Grainger, 2014; Romshoo et al., 2021; Liu et al., 2019; Luo et al., 2018a). Analysis of various modelling methods for BC particles showed that the selection of an appropriate size representation (polydisperse size method) and an appropriate morphological representation (aggregate morphology) could result in a more realistic prediction of $\sigma_{abs}$, SSA, and $MAC_{BC}$. Although optical simulations are time-consuming, it is suggested to use polydisperse size method for future modelling studies of BC fractal aggregates. The findings of this study are a good example of how parallel measurements and modelling research can reduce the uncertainties in optical properties of BC. It is also important to note that aerosols in the atmosphere contain a mixture of fresh, semi-aged, and aged aerosols (Fu et al., 2012) that will have either fractal or non-fractal morphology, depending on various factors. This study investigated the optical properties of BC particles with coatings up to 53%. It is recommended that further investigations be conducted on ambient or laboratory BC particles with coatings that exceed 50% to determine how the aggregate representation performs when particles are more aged. The long-term goal should be to incorporate the findings of such studies for black carbon parametrization scheme development and application to global climate models.

**Appendix A: Experimental setup and instrumentation**

Figure A1 shows an overview of the experimental setup used in experiment E1: measurements of thermally denuded nascent black carbon particles. The pre-treated particles were divided into four aerosol flows (i.e. sampling lines) and delivered to the different instruments. One part of the aerosol flow passed through a Mobility Particle Size Spectrometer (MPSS, TROPOS design; sample flow rate of 1 lpm; Wiedensohler et al., 2012; 2018a) which measured the particle number size distribution of the black carbon particles. Another part of the aerosol flowed was guided to a Cavity Attenuated Phase Shift Extinction monitor (CAPS $PM_{ex\ 630}$, Aerodyne Res. Inc., USA; flow rate of 1 lpm) which measured the light extinction coefficient, $\sigma_{ext}$ at wavelength of 630 nm. The other part of the aerosol flow entered an aethalometer (AE33 Aethalometer, Magee Scientific, Berkeley, USA; flow rate of 5 lpm) which monitored the equivalent black carbon concentration at seven wavelengths between 370 and 950 nm. The equivalent black carbon concentration was converted into the aerosol light absorption coefficient ($\sigma_{abs}$), as described in Müller et al. (2011). A further part of the aerosols was passed through a nephelometer (Aurora 4000, Ecotech Pvt Ltd, Melbourne, Australia) and a multi-angle absorption photometer (MAAP, type 5012, Thermo Scientific, Franklin, MA) running in tandem configuration at a flow rate of 10 lpm that measured the particle light scattering coefficient, $\sigma_{scat}$, and the absorption coefficients, $\sigma_{abs}$, respectively. The $\sigma_{abs}$ obtained from the AE33 was corrected by a factor of 0.95 to 1.3 to match the $\sigma_{abs}$ from MAAP. The $\sigma_{scat}$ measured from the nephelometer was also corrected for truncation errors due to the finite viewing angle of the detector, given in detail by Müller et al. (2009).

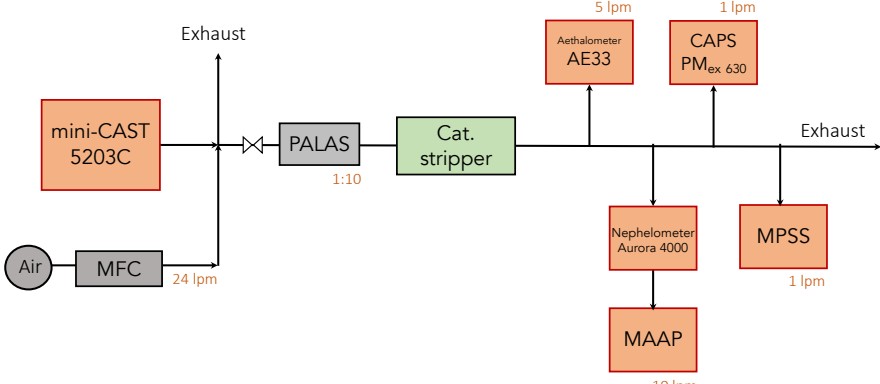

**Figure A1.** Experimental setup of E1: generation and measurements of denuded BC particles. The sootgenerator mini-CAST 5203C was used to generate the particles under different operating conditions given in Table 1. A mass flow controller (MFC) was used to mix the aerosols generated from mini-CAST 5203C with air, and then a Catalytic Stripper at 350°C was used to remove the volatile contents. The heated black carbon particles were

880 divided into four aerosol flows and delivered to instruments. The different instruments measuring the physical and optical properties are Mobility Particle Size Spectrometer (MPSS), Cavity Attenuated Phase Shift Extinction monitor (CAPS $PM_{ex\,630}$), aethalometer (AE33), nephelometer, and multi-angle absorption photometer (MAAP).

Figure A2 shows a schematic diagram of the experimental setup used in E2: measurements of untreated nascent

black carbon particles. The aerosol from the mini-CAST using a dilution system (PALAS VKL 10, PALAS, Karlsruhe, Germany) was fed into the mixing chamber and delivered to various measurement systems through several sampling ports. The aerosol from the first sampling port flowed at 6 lpm into a nephelometer (Aurora 4000) and a multi-angle absorption photometer (MAAP, type 5012) arranged in tandem configuration. The aerosol from a second port was guided to an Aethalometer (AE33; flow rate of 8 lpm), and three Cavity Attenuated Phase

Shift Extinction monitor: CAPS $PM_{ex\,450}$, CAPS $PM_{ex\,530}$, and CAPS $PM_{ex\,630}$ (flow rate of 8 lpm), which measured at wavelengths of 450, 530, and 630 nm, respectively. Subsequently, the aerosol flowed into the MPSS (TROPOS design; flow rate of 1 lpm) and the Cavity Attenuated Phase Shift single scatter albedo monitor (CAPS $PM_{SSA\,630}$, Aerodyne Res. Inc., USA; flow rate of 1lpm) from the third and fourth sampling port, respectively. The CAPS $PM_{SSA\,630}$ measured the scattering coefficient, $\sigma_{sca}$, and the extinction coefficient, $\sigma_{ext}$ at a wavelength of 630

895 nm. Through the fourth port, the aerosol mass concentration was determined by using the Tapered Element Oscillating Microbalance (TEOM 1405, Thermo Scientific, Franklin, MA; flow rate of 3 lpm). The aerosol from the last port was sampled on quartz fibre filters at a flow rate of 2-3 lpm and subsequently analysed by a EC/OC analyser (Sunset Laboratory Inc., Hillsborough, USA). The loaded quartz fibre filters were analysed at different laboratories, including METAS (Switzerland) and NPL (UK). For a better overview, the details of the

900 instrumentation used in E1 and E2 laboratory experiments are summarized in Table A1.

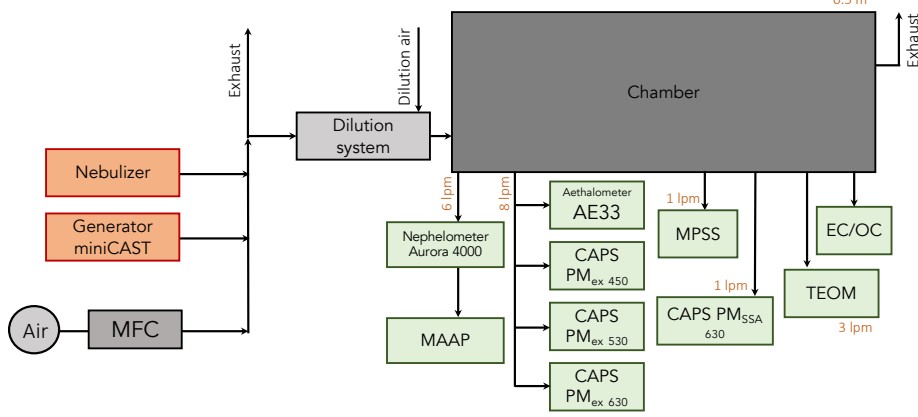

**Figure A2.** Experimental setup of E2: generation and measurements of BC particles with organics. The black

carbon particles are generated using different mini-CAST generators operated under flow settings given in Table

1. Mini-CAST sootgenerators produce aerosols that are mixed with air from the mass flow controller (MFC). After passing through a dilution system, the aerosols enter the mixing chamber. The black carbon particles are delivered to various instruments measuring physical, optical, and chemical properties. The instruments used in this experiment are the aethalometer (AE33), nephelometer, multi-angle absorption photometer (MAAP), Cavity Attenuated Phase Shift Extinction monitor (CAPS PM$_{ex\ 630}$), a Cavity Attenuated Phase Shift Extinction monitor (CAPS PM$_{ex\ 530}$), Cavity Attenuated Phase Shift Extinction monitor (CAPS PM$_{ex\ 450}$), Cavity Attenuated Phase Shift single scatter albedo monitor (CAPS PM$_{SSA\ 630}$), Mobility Particle Size Spectrometer (MPSS), and Tapered Element Oscillating Microbalance (TEOM).

**Table A1.** Details of the instruments used in E1 and E2.

| Instrument | Manufacturer | Function or measured variable | Experiment |
|---|---|---|---|
| mini-CAST 5203 Type C | Jing | Soot generator | E1, E2 |
| mini-CAST 5201 Type BC | Jing | Soot generator | E2 |
| mini-CAST 5303 Type C | Jing | Soot generator | E2 |
| Mobility Particle Size Spectrometer (MPSS) | TROPOS | Particle number size distribution (Mobility diameter) | E1, E2 |
| Cavity Attenuation Phase Shift Spectrometer (CAPS PM$_{ex\ 630}$) | Aerodyne Research | Particle light extinction coefficients ($\sigma_{ext}$) in Mm$^{-1}$ at $\lambda$ = 630 nm | E1, E2 |
| Cavity Attenuation Phase Shift Spectrometer (CAPS PM$_{ex\ 530}$ | Aerodyne Research | Particle light extinction coefficients ($\sigma_{ext}$) in Mm$^{-1}$ at $\lambda$ = 530 nm | E2 |
| Cavity Attenuation Phase Shift Spectrometer (CAPS PM$_{ex\ 450}$ | Aerodyne Research | Particle light extinction coefficients ($\sigma_{ext}$) in Mm$^{-1}$ at $\lambda$ = 450 nm | E2 |
| Cavity Attenuation Phase Shift Spectrometer (CAPS PM$_{ssa\ 630}$ | Aerodyne Research | Particle light scattering and extinction coefficients at Mm$^{-1}$ at $\lambda$ = 630 nm | E2 |
| Aethalometer AE33 | Magee Scientific | Particle light absorption coefficients ($\sigma_{abs}$) in Mm$^{-1}$ at seven wavelength, $\lambda$ = 370, 470, 520, 590, 660, 880, and 950 nm | E1, E2 |
| Multi-angle absorption photometer (MAAP) | Thermo-Scientific | Particle light absorption coefficients ($\sigma_{abs}$) in Mm$^{-1}$ at 637 nm | E1, E2 |
| Tapered Element Oscillating Microbalance (TEOM) | Thermo-Scientific | Particle mass concentration | E2 |
| Nephelometer | Aurora | Particle light scattering coefficients ($\sigma_{sca}$) in Mm$^{-1}$ at 635 nm | E1, E2 |

**Appendix B: Details about modelling**

The first method for calculation of number of primary particles per aggregate ($N_{pp}$) from the $d_{p,\bar{N}}$ (Rissler et al. 2012; Bladh et al. 2001) is given as:

$$N_{pp}\left(d_{p,\bar{N}}\right) = \frac{m_{agg}\left(d_{p,\bar{N}}\right)}{m_{pp}\left(d_{p,\bar{N}}\right)} = \frac{\frac{d_{p,\bar{N}}}{2}^3 \cdot \rho_{eff}}{R_{pp}^3 \cdot \rho_{pp}},$$
(B1)

where the mass of the aggregate $m_{agg}$ was assumed to have an effective density $\rho_{eff}$ (g/cm$^3$), and the mass of the primary particle is $m_{pp}$ was assumed to have a density $\rho_{pp}$ of 1.8 g/cm$^3$ (Rissler et al. 2013). Following the study of Malik et al. 2011, the $\rho_{eff}$ was assumed as 0.76 ± 0.04 g/cm$^3$ for $d_{p,\bar{N}}$ < 50nm, and for 250 < $d_{p,\bar{N}}$ < 50 nm, the $\rho_{eff}$ was 0.51 ± 0.04 g/cm$^3$.

The second method developed by Sorensen (2011) is applicable to black carbon fractal aggregates since they are formed by the Diffusion Limited Aggregation DLA process and fall under the slip regime. Slip regime is a transition between the continuum and free molecular regime, where the Knudsen number $Kn$ lies between 0.1 to 10. The Knudsen number $Kn$ is the ratio of the molecular free path to the aggregate mobility radius (Friedlander 1977). The conversion is given as:

$$d_{p,\bar{N}} = 2R_{pp} \cdot N_{pp}^{0.46} \quad N_{pp} < 100 \,, \tag{B2}$$
$$d_{p,\bar{N}} = 2R_{pp} \cdot (10^{-2x+0.92}) \cdot N_{pp}^{x} \quad N_{pp} > 100 \,, \tag{B3}$$

with a mobility mass scaling exponent of $x = 0.51 Kn^{0.043}$ for $0.46 < x < 0.56$. In this study, the average value of the mobility mass scaling exponent $x = 0.51 \pm 0.02$ was assumed.

The third method, developed by Schmidt-Ott. (1988) follows a power law function, and is given as:

$$N_{pp} = K \cdot \left(\frac{d_{p,\bar{N}}}{2R_{pp}}\right)^{D_{fm}}, \tag{B4}$$

where, $K$ is a pre-factor, $D_f$ is the fractal dimension, and $D_{fm}$ is the mass mobility exponent. According to Park, Kittelson, & McMurry (2004) the relation between $D_{fm}$ and $D_f$ is $D_{fm} = 1.26 \cdot D_f$ for diesel soot, which was also used in this study. The value of $D_f$ was taken from literature. For all the three conversion methods, the $N_{pp}$ was estimated using both the number mean mobility diameter ($d_{p,\bar{N}}$), and the volume mean mobility diameter ($d_{p,\bar{V}}$).

**Table B1.** Values of $m_{re}$ and $m_{im}$ used in this study (Kim et al., 2015) for elemental carbon (EC) and organic carbon (OC).

| Refractive index ($m$) | Wavelength ($nm$) | | |
|---|---|---|---|
| | 467 | 530 | 660 |
| **Elemental carbon (EC)** | | | |
| $m_{re}$ | 1.92 | 1.96 | 2.0 |
| $m_{im}$ | 0.67 | 0.65 | 0.63 |
| | | | |
| **Organic carbon (OC)** | | | |
| $m_{re}$ | 1.59 | 1.47 | 1.47 |
| $m_{im}$ | 0.11 | 0.04 | 0 |

## Appendix C: Symbols and acronyms

**Table C1.** Symbols used.

| Symbol | Meaning |
|---|---|
| $\sigma_{ext}$ | Extinction coefficient |
| $\sigma_{abs}$ | Absorption coefficient |
| $\sigma_{sca}$ | Scattering coefficient |
| $Q_{ext}$ | Extinction efficiency |
| $Q_{abs}$ | Absorption efficiency |
| $Q_{sca}$ | Scattering efficiency |
| $g$ | Asymmetry parameter |
| $m_{re}$ | Real part of refractive index |
| $m_{im}$ | Imaginary part of refractive index |
| $D_f$ | Fractal dimension |
| $N_{pp}$ | Number of primary particles in aggregate |
| $a_{pp}$ | Radius of a primary particle (no coating) |
| $a_{in}$ | Inner radius of a primary particle (with coating) |
| $a_o$ | Outer radius of a primary particle (with coating) |
| $D_{in}$ | Inner diameter of volume equivalent sphere |
| $D_o$ | Outer diameter of volume equivalent sphere |
| $d_p$ | Mobility diameter |
| $d_{p,\bar{N}}$ | Number mean mobility diameter |
| $d_{p,\bar{V}}$ | Volume mean mobility diameter |
| $\phi$ | Flame equivalence ratio |
| $f_{oc}$ | Fraction of organic carbon |
| $D_i$ | Diameter of $i^{th}$ SMPS size bin |
| $n_i$ | Number concentration of $i^{th}$ SMPS size bin |
| $Q_{abs\_i}$ | Absorption efficiency of $i^{th}$ SMPS size bin |

| | |
|---|---|
| $C_{abs\_i}$ | Absorption cross-section of i[th] SMPS size bin |

**Table C2.** Acronyms used.

| Acronym | Meaning |
|---|---|
| **BC** | Black carbon |
| **SSA** | Single scattering albedo |
| $MAC_{BC}$ | Mass absorption cross-section |
| **AAE** | Ångström absorption exponent |

**Code availability**

The software used to generate the fractal aggregates is available at t https://sites.google.com/view/fabriceonofri/ aggregates/fractal-like-aggregates-diffusion-model (Wózniak and Onofri, 2022). The code used to model the optical properties of fractal aggregate, the multi-sphere T-matrix (MSTM) is publicly available at https://eng.auburn.edu/users/dmckwski/scatcodes/ (Mackowski, 2022)

**Data availability**

The data obtained from this study are available upon request from the corresponding author (baseerat@tropos.de).

**Author contributions**

TM and BR designed the study, with assistance from MP and AW. The first laboratory experiments and analysis were conducted by JS and AN. The second laboratory experiment was conducted by TM, SP, JS, BR, AN, KV, MNE, KC, PQ, MG, KE, CR, and FGL. The modelling experiments were carried out by BR, with help from TM. The paper was written by BR and reviewed, commented on, and edited by TM, MP, KV, MNE, AW, JS, AN, MG, KE, KC and PQ. The analysis of the filter samples were conducted by KV, MNE, CR and KC. The TEM images before and after the catalytic stripper were provided by KV and MNE.

**Competing interests**

The authors declare that they have no conflict of interest.

**Acknowledgement**

This work is supported by the 16ENV02 Black Carbon project of the European Union through the European Metrology Programme for Innovation and Research (EMPIR). We thank the editor, Pierre Herckes, as well as the two anonymous referees for their insightful suggestions, which enabled us to improve the paper.

**Financial support**

This research has been supported by the "Metrology for light absorption by atmospheric aerosols" project funded by the European Metrology Programme for Innovation and Research (EMPIR, grant no. 16ENV02 Black Carbon).

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
