# Peer review of "Importance of size representation and morphology in modelling optical properties of black carbon: comparison between laboratory measurements and model simulations"

_EGUsphere, 2022_

## Referee Comment (RC1)

Reviewer comment on

**Importance of size representation and morphology in modelling optical properties of black carbon: comparison between laboratory measurements and model simulations**

General comments:

The manuscript compares the optical properties of laboratory generated soot particled with those obtained from model calculations. The paper is generally well written and worth to be published after minor modifications.

The authors run model calculations and perform laboratory measurements for bare (or half-bare) soot particles and for soot particles with organics. These two types of particles were generated in two separate experiments E1 and E2.

In the results section, several optical parameters obtained from the model runs are discussed and compared with the measured values. It would be useful for the readers, if the same optical parameters (including the concentration dependent parameters :absorption and scattering coefficient) would have been discussed for bare and undenuded soot particles. For example AAE from the 2nd experiment is not mentioned and MAC is discussed only for the E2 experiment.

Although the authors mention in the paper that the resmallader should keep in mind that for smaller particles the Catalytic Stripper was less effective, thus the particles can not be considered bare soot particles this introduces some uncertainty in the comparison of model and measured data. Unfortunately there is no EC/TC measurements for denuded particles, but the authors should give an estimation for the uncertainty of the measured data caused by the residue of the organic material. Fig. S2 indicates that the modeled absorption coefficient of the fractal bare BC using polydisperse method is smaller for larger particles (160 nm) than the experimentally determined value. Might this indicate the presence of organic residue for larger particles?

Specific comments:

Line 311: the word modeled should be modelled

Line 356: the density is not mentioned in the equation

The yellow band of the measured data in figure 7, 8 and 9 is hardly visible.

Figure S1: the "blue stars" appear green because of the green edge of the symbol.

For which wavelength was the MAC calculated?

---

## Author Comment (AC1)

Reviewer #1 (R1):

We would like to thank the reviewer for the insightful review and constructive comments on our manuscript. The time taken by the Reviewer to review and evaluate the manuscript is highly appreciated. We have considered all comments and suggestions and incorporated them into the revised manuscript which have improved the quality of the revised manuscript. The point by point response to all the comments and suggestions of reviewer #1 (R1) is provided in the following sections. For clarity, the reviewer's comments are provided in blue, the author's response (AR) is in black, and the revised parts of the manuscript are highlighted in red in the revised manuscript.

General comments (GC):

R1 GC1: The manuscript compares the optical properties of laboratory generated soot particles with those obtained from model calculations. The paper is generally well written and worth to be published after minor modifications.

The authors run model calculations and perform laboratory measurements for bare (or half-bare) soot particles and for soot particles with organics. These two types of particles were generated in two separate experiments E1 and E2.

**AR:** The authors thank the reviewer for the constructive general remarks.

R1 GC2: In the results section, several optical parameters obtained from the model runs are discussed and compared with the measured values. It would be useful for the readers, if the same optical parameters (including the concentration dependent parameters :absorption and scattering coefficient) would have been discussed for bare and undenuded soot particles. For example AAE from the 2nd experiment is not mentioned and MAC is discussed only for the E2 experiment.

**AR:** We thank the reviewer for the comment. We agree with the reviewer that it is important to discuss and make these results available to readers for their better understanding and interpretation. As suggested, the mass absorption cross-section ($MAC_{BC}$) was calculated for the four cases of the experiment E1. Modelled values of $MAC_{BC}$ ranged from 2.44 to 4.66 m$^2$/g when using pure BC. Because of the unavailability of an instrument directly measuring the mass in E1, the mass was calculated assuming a density of 1.8 g cm$^{-3}$ (Park et al., 2004). In smaller BC particles ($d_{p,\bar{V}}$ of 60 and 106 nm), the modelled $MAC_{BC}$ is larger than the measured $MAC_{BC}$. This may be because of the reason that the smaller particles contain higher residual organic matter (Zhang et al., 2020), which results in an underestimation of the measured $MAC_{BC}$ when a density of 1.8 g cm$^{-3}$ is used.

Additionally, the absorption ångström exponent (AAE) was calculated for the experiment E2. As also shown in experiment E1, the modelled AAE values matched the measurements more closely when a spherical representation for BC is used.

The above mentioned results have been incorporated in the revised manuscript as supplementary material and will be also graphically represented as shown below:

[Figure]

[Figure]

R1 GC3: Although the authors mention in the paper that the resmallader should keep in mind that for smaller particles the Catalytic Stripper was less effective, thus the particles cannot be considered bare soot particles this introduces some uncertainty in the comparison of model and measured data. Unfortunately there is no EC/TC measurements for denuded particles, but the authors should give an estimation for the uncertainty of the measured data caused by the residue of the organic material.

**AR:** We thank the reviewer for this comment. The efficiency of the Catalytic Stripper depends on the volatility of the organic matter present in the aerosol particles. The uncertainty associated with the Catalytic Stripper in removing the organic matter was studied by Mamakos et al., 2013. They reported that in the 21–250◦C temperature range, the Catalytic Stripper is able to remove up to 96% of the more volatile fraction of organic matter. However, the Catalytic Stripper removes 30–60% of the less volatile organic matter in the 250–500◦C temperature range. For experiment E1 of our study, we modelled the optical properties of particles passing through a Catalytic Stripper at 350°C, so we may expect that 40-70% of less volatile organic matter residues will still be present.
The future possibility of having EC/TC measurements in such an experiment will be useful to report the absolute uncertainty in terms of residual organic matter when using a Catalytic Stripper with optical instruments.

The above points have been summarized under the methods section of the revised manuscript as follows:

For modelling the particles from the denuding experiment E1, the simulated particles are assumed to be bare black carbon, since a Catalytic Stripper was used to remove the volatile organic matter. Some residuals, however, are left behind by the Catalytic Strippers, depending on the volatility of the organic matter. Mamakos et al. (2013) reported that in the 21–250◦C temperature range, the Catalytic Stripper is able to remove up to 96% of the more volatile fraction of organic matter. However, in the 250–500◦C temperature range, the Catalytic Stripper removes 30–60% of the less volatile organic matter. This must be noted when comparing the modelled optical results with their equivalent laboratory measurements.

Further, the following relevant text was added in the discussion section of the revised manuscript:

Furthermore, smaller particles contain more organic content than larger ones (Zhang et al., 2020), leading to a less effective removal by Catalytic Stripper. In case the smaller particles were immature solid soot with embedded organic content, the assumption that they are bare may account for the underestimation of the modelled AAE in comparison to the measured values. In future, it will be useful to have EC/TC measurements in such an experiment in order to determine the absolute uncertainty in terms of the residual organic matter when the Stripper is used with optical instruments.

R1 GC4: Fig. S2 indicates that the modeled absorption coefficient of the fractal bare BC using polydisperse method is smaller for larger particles (160 nm) than the experimentally determined value. Might this indicate the presence of organic residue for larger particles?

AR: We thank the reviewer for highlighting this point. In the Fig. S2, in the aggregate representation, we had by mistake shown the results for all the other three operating points of Catalytic Stripper. It is for this reason that the modelled results using the aggregate were shown lower than the measured values. It has been corrected in order to display the modelled results at 350°C condition (BC particles pass through the Catalytic Stripper at 350°C). In all sized particles, the modelled absorption coefficient ($\sigma_{abs}$) matches the measured $\sigma_{abs}$ when using an aggregate morphological representation. However, in larger particle (> 150 nm), the accuracy in the modelled $\sigma_{abs}$ was comparatively higher. This is because in larger particles, the organic matter residues are lower, making the assumption of pure BC more suitable. This can also be seen in the results of modelled AAE, where the discrepancy is higher for smaller particles indicating the presence of more residual organic matter in smaller particles.

[Figure]

The above figure is updated in the revised version of the manuscript.

Specific comments:

1) Line 311: the word modeled should be modelled

    AR: Thank you for the correction. The change has been made in the revised manuscript.

2) Line 356: the density is not mentioned in the equation

    AR: Thank you for the correction. The density is included in eq. (6) in the revised manuscript as follows:

$$MAC_{BC} = \frac{C_{abs}}{m_{BC}} = \frac{C_{abs}}{\frac{1}{6}\pi d^3 \cdot \rho_{BC}} \, , \tag{6}$$

    where $\rho_{BC}$ is the density of black carbon and taken in this study to be 1.8 g cm$^{-3}$ (Park et al., 2004).

3) The yellow band of the measured data in figure 7, 8 and 9 is hardly visible.

    AR: Thank you for highlighting this point. The measured data is highlighted better for Fig. 7 to Fig. 9 in the revised manuscript.

4) Figure S1: the "blue stars" appear green because of the green edge of the symbol.

**AR:** Thank you for highlighting this point. For better visibility, the color of the star is changed to green.

5) For which wavelength was the MAC calculated?

**AR:** The $MAC_{BC}$ was calculated at 660 nm. The following sentence has been added in the revised manuscript:

The $MAC_{BC}$ was calculated using the $C_{abs}$ at a wavelength of 660 nm.

---

## Author Comment (AC2)

Reviewer #2 (R2):

We would like to thank the reviewer for providing an insightful review and valuable comments. It is much appreciated that you took the time to review the manuscript. We have taken into consideration all of your comments/suggestions, and incorporated them into the revised manuscript, which has enhanced the quality of the revised manuscript. The point by point response to all the comments and suggestions of reviewer #2 (R2) is provided in the following sections For clarity, the reviewer's comments are provided in blue, the author's response (AR) is in black, and the revised parts of the manuscript are shown in red

**R2:** In this paper, the authors describe various models of the optical properties of black carbon particles and compare them with laboratory measured values for soot generated with a mini-CAST. The work done is quite extensive and as such, it is important to disseminate it to the community. However, I think the paper could significantly be improved in terms of clarity and readability and would benefit from providing a clearer and more succinct overall message.

**AR:** The authors thank the reviewer for the constructive general remarks. The incorporation of the comments and suggestions of the reviewer have significantly improved the quality of the manuscript.

**General comments (GC):**

**R2 GC1:** This study compares models with laboratory measurements only. However, black particles in the atmosphere have been shown to be much more complex than those generated in the laboratory, for example, those emitted during biomass burning or those transported very far from the source. It is reasonable and already impressive to focus only on laboratory-generated particles. Still, the conclusions might need to be put more in view of this limitation/caveat to avoid the impression that the implications of the findings on describing the actual properties of atmospheric aerosols might be overinflated.

**AR:** We thank the reviewer for this comment and agree that it is important to keep in mind the complex ageing process of BC in the atmosphere. As described in the introduction, BC particles have a fractal aggregate morphology. When the BC fractal particles remain in the atmosphere for a longer period of time, they are transformed into coated compact particles. According to some studies, TEM images of aerosols in the atmosphere showed an intact fractal aggregate morphology surrounded by a coating (Fu et al., 2011). However, it is still uncertain how much portion of the BC containing aerosols particles in the atmosphere retain their fractal structure. Whether such semi-aged particles are found in the atmosphere depends on the emission sources, atmospheric chemistry, and meteorology at the site. Therefore, aerosols in the atmosphere are expected to contain a combination of fresh, semi-aged, and aged aerosols with or without fractal morphologies, depending on various factors.
However, laboratory-produced particles have a shorter residence time and therefore are subject to limited ageing. In our case, we had coating up to 53%, in which using the fractal aggregate morphology performed well for $\sigma_{abs}$ and SSA. It is recommended that further investigations be conducted on ambient or laboratory BC containing particles with coatings that exceed 50% to determine how the fractal aggregate morphology performs when particles are more aged.

The conclusion section of the revised manuscript has been accordingly rephrased and modified to summarize the above discussion:

This study provides experimental support for previous theoretical work based on BC as fractal aggregates (e.g., Kahnert, 2010; Adaichi et al., 2010; Kahnert and Kanngießer, 2020; Smith and Grainger, 2014; Romshoo et al., 2021; Liu et al., 2019; Luo et al., 2018). Analysis of various modelling methods for BC particles showed that the selection of an appropriate size representation (polydisperse size method) and an appropriate morphological representation (aggregate morphology) could result in a more realistic prediction of BC's optical properties ($\sigma_{abs}$ and SSA). Although optical simulations are time-consuming, it is suggested to use polydisperse size method for future modelling studies of BC fractal aggregates. The findings of this study are a good example of how parallel measurements and modelling can reduce the uncertainties in optical properties of BC. It is also important to note that aerosols in the atmosphere contain a mixture of fresh, semi-aged, and aged aerosols (Fu et al., 2012) that will have either fractal or non-fractal morphology, depending on various factors. This study investigated the optical properties of BC particles with coatings up to 53%. It is recommended that further investigations be conducted on ambient or laboratory BC particles with coatings that exceed 50% to determine how the aggregate representation performs when particles are more aged. The long-term goal should be to incorporate the findings of such studies for black carbon parametrization scheme development and application to global climate models.

Fu, H., Zhang, M., Li, W., Chen, J., Wang, L., Quan, X., and Wang, W.: Morphology, composition and mixing state of individual carbonaceous aerosol in urban Shanghai, Atmos. Chem. Phys., 12, 693–707, https://doi.org/10.5194/acp-12-693-2012, 2012.

**R2 GC2:** The authors switch frequently (but mostly from the first part of the paper to the second) between the term "soot" and the term "black carbon". While this is understandable, this can cause confusion in view especially of a few papers discussing this terminology issue – an issue that can be problematic and is partially still unresolved (e.g., Buseck, P. R.; Adachi, K.; Gelencsér, A.; Tompa, É.; Pósfai, M., ns-Soot: A Material-Based Term for Strongly Light-Absorbing Carbonaceous Particles. Aerosol Science and Technology 2014, 48, (7), 777-788. And Petzold, A.; Ogren, J. A.; Fiebig, M.; Laj, P.; Li, S. M.; Baltensperger, U.; Holzer-Popp, T.; Kinne, S.; Pappalardo, G.; Sugimoto, N.; Wehrli, C.; Wiedensohler, A.; Zhang, X. Y., Recommendations for reporting "black carbon" measurements. Atmospheric Chemistry and Physics 2013, 13, (16), 8365-8379.)

**AR:** We thank the reviewer for pointing out the discrepancy in terminology used. In agreement with the reviewer, we will use "black carbon" throughout the manuscript. The changes have been accordingly made in the revised version of the manuscript.

**R2 GC3:** A bit related to the first point, how representative is the Mini-CAST black carbon of ambient soot? Even for bare black carbon?

**AR:** We thank the reviewer for bringing this point forward. According to TEM analysis, BC particles emitted from a Mini-CAST burner have fractal morphologies (Ouf et al., 2016). Depending on the burning conditions of the flame as described by the flame equivalence ratio ($\phi$), the percentage of organic matter which forms an external layer around the BC may vary (Mamakos et al., 2018). Ouf et al., 2016 showed images of how the structure of the BC aggregate varied with the operating conditions of the mini-CAST burner. They operated the mini-CAST burner on three conditions producing BC aggregates with $f_{oc}$ of 4% (CAST1), 47% (CAST2), and 87% (CAST3). The TEM images from the three cases are shown below.

[Figure]

In our study, we produced BC particles with $f_{oc}$ up to 53%. Therefore, we expect the BC particles generated in our to resemble the CAST2 case of Ouf et al., 2016. The BC particles are expected to be fractal in morphology with a coating around them.

When we study BC containing aerosols in the atmosphere, we find that, in addition to the nature of the emission source, factors such as the residence time and atmospheric chemistry have an impact on their morphology. During the same period, multiple kinds of BC-containing particles with differing levels of aging can exist simultaneously. Fu et al., 2011 presented the following image which illustrates different types of BC-containing particles collected from Shanghai's atmosphere. As seen in the picture below, BC can maintain fractal aggregate morphology even as it ages.

[Figure]

This means that the particles produced by mini-CAST burners are not representative of all atmospheric BC particles, but only a portion of them that have not significantly aged. In order to study heavily coated BC with more than 50% coating, the mini-CAST must be operated in specific conditions as described in Ouf et al., 2016.

The following lines in red have been added to section 2.2.1 of the revised manuscript:

In order to model the optical properties of soot, it is important to choose the most appropriate morphological representation for soot particle. This step is considered particularly important because the modelled optical properties were further validated with the measurements from E1 and E2. TEM images were not available for this study; therefore, the morphological representations of soot were selected based on TEM images from the previous laboratory studies using the mini-CAST generators (Ess et al., 2021; Ouf et al., 2016). In the mini-CAST generator, BC particles produced have fractal morphologies, with varying amounts of organics attached to the edges, without altering the inner structure of the core (Ouf et al., 2016). In the radiative modelling studies, it is possible to simulate this laboratory result by assuming a sphere of coating around each individual primary particle of a BC aggregate (Luo et al., 2018). In addition to the TEM images from Ess et al. (2021), the operating conditions of the mini-CAST burners during experiments E1 and E2 (Table. 1), and the fraction of organic carbon of soot particles from E2 were also kept in mind while selecting the morphological representations. Additional discussion of the TEM images of BC particles taken from mini-CAST and atmosphere can be found in the Supplementary material.

Ouf, F. X., Parent, P., Laffon, C., Marhaba, I., Ferry, D., Marcillaud, B., Antonsson, E., Benkoula, S., Liu, X. J., Nicolas, C., Robert, E., Patanen, M., Barreda, F. A., Sublemontier, O., Coppalle, A., Yon, J., Miserque, F., Mostefaoui, T., Regier, T. Z., Mitchell, J. B. A. and Miron, C.: First in-flight synchrotron X-ray absorption and photoemission study of carbon soot nanoparticles, Scientific Reports, 6, doi:10.1038/srep36495, 2016.

Furthermore, the following lines in red have been added in the Introduction section of the revised manuscript:

High-resolution transmission electron microscopy (TEM) analysis of BC samples from ambient and laboratory studies revealed that BC particles comprise agglomerates made from numerous graphitic soot spherules (Betrancourt et al., 2017; Gini et al., 2016). Over time, BC agglomerates undergo complex changes in their size, morphology, and composition, depending on post-emission atmospheric conditions (Fierce et al., 2015). TEM images from Shanghai's atmosphere presented by Fu et al. (2011) showed a variety of BC-containing particles at various stages of aging, of which some semi-aged particles retained fractal aggregate morphology. The BC particles are often found together with other combustion by-products such as organic matter, which enhance the particle light absorption through the lensing effect (Fuller et al., 1999). With increasing residence time of BC in the atmosphere, an aging process occurs, leading to a growth of BC agglomerates into much more compact

structures. This is mainly due to the formation of coatings and hygroscopic properties (Petzold et al., 2005; Bond et al., 2006; Abel et al., 2003).

The above discussion along with the pictures will be provided as Supplementary material in the revised manuscript.

Fu, H., Zhang, M., Li, W., Chen, J., Wang, L., Quan, X., and Wang, W.: Morphology, composition and mixing state of individual carbonaceous aerosol in urban Shanghai, Atmos. Chem. Phys., 12, 693–707, https://doi.org/10.5194/acp-12-693-2012, 2012.

**R2 GC4:** The references used sometimes are a bit limited and maybe biased.

**AR:** The comment made by the reviewer is acknowledged, making the following addition of references in the revised manuscript.

Calcote, H. F.: Mechanisms of soot nucleation in flames – A critical review, Combust. Flame, 42, 215–242, https://doi.org/10.1016/0010-2180(81)90159-0, 1981.

Cappa, C. D., Onasch, T. B., Massoli, P., Worsnop, D. R., Bates, T. S., Cross, E. S., Davidovits, P., Hakala, J., Hayden, K. L., Jobson, B. T., Kolesar, K. R., Lack, D. A., Lerner, B. M., Li, S. M., Mellon, D., Nuaaman, I., Olfert, J. S., Petäjä, T., Quinn, P. K., Song, C., Subramanian, R., Williams, E. J., and Zaveri, R. A.: Radiative absorption enhancements due to the mixing state of atmospheric black carbon, Science, 337, 1078–1081, https://doi.org/10.1126/science.1223447, 2012

Liu, D. T., Whitehead, J., Alfarra, M. R., Reyes-Villegas, E., Spracklen, D. V., Reddington, C. L., Kong, S. F., Williams, P. I., Ting, Y. C., Haslett, S., Taylor, J. W., Flynn, M. J., Morgan, W. T., McFiggans, G., Coe, H., and Allan, J. D.: Black-carbon absorption enhancement in the atmosphere determined by particle mixing state, Nat. Geosci., 10, 184–132, https://doi.org/10.1038/ngeo2901, 2017

Wu, Y., Cheng, T., Zheng, L., and Chen, H.: Models for the radiative simulations of fractal aggregated soot particles thinly coated with non-absorbing aerosols, J. Quant. Spectrosc. Radiat. Transf., 182, 1–11, https://doi.org/10.1016/j.jqsrt.2016.05.011, 2016.

Fu, H., Zhang, M., Li, W., Chen, J., Wang, L., Quan, X., and Wang, W.: Morphology, composition and mixing state of individual carbonaceous aerosol in urban Shanghai, Atmos. Chem. Phys., 12, 693–707, https://doi.org/10.5194/acp-12-693-2012, 2012.

Ouf, F. X., Parent, P., Laffon, C., Marhaba, I., Ferry, D., Marcillaud, B., Antonsson, E., Benkoula, S., Liu, X. J., Nicolas, C., Robert, E., Patanen, M., Barreda, F. A., Sublemontier, O., Coppalle, A., Yon, J., Miserque, F., Mostefaoui, T., Regier, T. Z., Mitchell, J. B. A. and Miron, C.: First in-flight synchrotron X-ray absorption and photoemission study of carbon soot nanoparticles, Scientific Reports, 6, doi:10.1038/srep36495, 2016.

Bhandari, J., China, S., Chandrakar, K. K., Kinney, G., Cantrell, W., Shaw, R. A., Mazzoleni, L. R., Girotto, G., Sharma, N., Gorkowski, K., Gilardoni, S., Decesari, S., Facchini, M. C., Zanca, N., Pavese, G., Esposito, F., Dubey, M. K., Aiken, A. C., Chakrabarty, R. K., Moosmüller, H., Onasch, T. B., Zaveri, R. A., Scarnato, B. V., Fialho, P. and Mazzoleni, C.: Extensive Soot Compaction by Cloud Processing from Laboratory and Field Observations, Scientific Reports, 9(1), 1–12, doi:10.1038/s41598-019-48143-y, 2019.

Scarnato, B. V.; Vahidinia, S.; Richard, D. T.; Kirchstetter, T. W., Effects of internal mixing and aggregate morphology on optical properties of black carbon using a discrete dipole approximation model. Atmospheric Chemistry and Physics 2013, 13, (10), 5089-5101.

Luo, J., Zhang, Y., Zhang, Q., Wang, F., Liu, J., and Wang, J.: Sensitivity analysis of morphology on optical properties of soot aerosols, Opt. Express, 26, A420–A432, https://doi.org/10.1364/oe.26.00a420, 2018b.

Kahnert, M.: Optical properties of black carbon aerosols encapsulated in a shell of sulfate: comparison of the closed cell model with a coated aggregate model, Opt. Express, 25, 24579–24593 https://doi.org/10.1364/oe.25.024579, 2017.

Tian, K., Thomson, K., Liu, F., Snelling, D., Smallwood, G., and Wang, D.: Determination of the morphology of soot aggregates using the relative optical density method for the analysis of tem images, Combust. Flame, 144, 782–791, doi:10.1016/j.combustflame.2005.06.017, 2006.

Luo, J., Zhang, Y. and Zhang, Q.: The Ångström exponent and single-scattering albedo of black carbon: Effects of different coating materials, Atmosphere, 11(10), 1103, doi:10.3390/atmos11101103, 2020.

Yuan, C., Zheng, J., Ma, Y., Jiang, Y., Li, Y. and Wang, Z.: Significant restructuring and light absorption enhancement of black carbon particles by ammonium nitrate coating, Environmental Pollution, 262, 114172, doi:10.1016/j.envpol.2020.114172, 2020.

**R2 GC5:** Could removing the volatile organic compounds with the thermos-stripper change the morphology of the black carbon particles? For example, would the surface tension acting on the monomers while the coating is evaporating/, or the heating cause partial collapse of the aggregate? The authors should comment on this potential issue.

**AR:** We thank the reviewer for bringing this point forward. Our partners from the project at the Federal Institute of Metrology METAS, Switzerland took some TEM measurements of BC particles from the miniCAST before and after the Catalytic Stripper. The tests were performed with 60 nm particles which typically have 20-25% OC/TC mass fraction. The images of the BC particles before and after the Catalytic Stripper are given below.

[Figure]

It was shown that there is no major morphological transformation after the BC particles are treated by the Catalytic Stripper. The BC particles remain fractal-like, and do not collapse to a compact structure.

The above discussion about the above facts along with the pictures will be provided in the Supplementary material of the revised manuscript.

**R2 GC6:** The conclusions seem to underline the need to account for polydispersity and accurate coating representation to improve the calculations of the optical properties of laboratory-generated black carbon, but the authors seem to gloss over some of the results previously discussed in the paper, for example, for the MAC of smaller particles where some of the spherical simulations seem better. So, I was left with a few mixed feelings about what is really the best representation.

**AR:** We thank the reviewer for the suggestion. We agree with the reviewer and have accordingly improved the conclusion to account for all the findings of the study in a balanced way.

In both the experiments E1 and E2, it was seen that the aggregate representation of BC performed well for modelling the $\sigma_{abs}$ and SSA. However, in contrast to the results of $\sigma_{abs}$ and SSA, the AAE modelled using the aggregate representation varied from the measured AAE by a factor of 1.5. In the case of larger particles ($\geq$100 nm), the modelled AAE from the sphere representation were in better agreement with the measured results. For smaller BC particles, both the sphere and aggregate representations underestimated the AAE. Similarly, for $MAC_{BC}$, both spherical and aggregate representation underestimated the $MAC_{BC}$ for smaller particles, though, the homogenous sphere representation was comparatively closer to the measured $MAC_{BC}$. For particles larger than 100 nm, the $MAC_{BC}$ was modelled more accurately when using the aggregate representation.

So overall, the aggregate representation performed well for modelling the $\sigma_{abs}$, SSA, and $MAC_{BC}$ for laboratory generated BC particles with $f_{oc}$ less than 53% and $d_{p,V}$ larger than100 nm. Whereas, the spherical representation performed well for modelling the AAE in larger particles ($d_{p,V}$ > 100 nm). However, for smaller particles, using both aggregate or spherical representation results in a larger discrepancy when modelling the AAE or $MAC_{BC}$. The discrepancy was more pronounced in the cases of experiments E1, where the EC/TC analysis was not conducted. Therefore, the discrepancy could be a result of the presence of organic matter in smaller particles even after being heated by the Catalytic Stripper. The presence of larger percentage of organic matter in smaller particles is also observed from the results of the EC/TC analysis of experiment E2 where the largest $f_{oc}$ was observed in case VIII with the smallest $d_{p,V}$ of 86 nm ($d_{p,N}$ = 48 nm).

The above discussion has been incorporated/modified in the Conclusion section of the revised manuscript.

In order to study the modelling approaches for coated BC particles, three kinds of mini-CAST soot generators were used to produce soot particles with organic carbon content between 35 - 65%. Four kinds of morphological representations for coated BC (two each for spherical and aggregate) were compared using both monodisperse and polydisperse particles. In the most of the results, the modelled SSA using the "coated aggregate" and "aggregate and sphere" representation was in good agreement with the measured SSA. Though it is less likely that laboratory-generated soot will resemble the "aggregate and sphere" representation, it can still be used when the coating only makes up a small part of the total particle volume. Moreover, when polydisperse method is used the accuracy improves by up to a factor of 2. Similar to E1, for coated BC also, the modelled AAE had larger discrepancies when using an aggregate representation, but matched the measured AAE when using a spherical assumption. Similarly, for $MAC_{BC}$, both spherical and aggregate representation underestimated the $MAC_{BC}$ for smaller particles, though, the homogenous sphere representation was comparatively closer to the measured $MAC_{BC}$. For particles larger than 100 nm, the $MAC_{BC}$ was modelled more accurately when using the aggregate representation.

So overall, the aggregate representation performed well for modelling the $\sigma_{abs}$, SSA, and $MAC_{BC}$ for laboratory generated BC particles with $f_{oc}$ less than 53% and $d_{p,V}$ larger than100 nm. Whereas, the spherical representation performed well for modelling the AAE in larger particles ($d_{p,V}$ > 100 nm). However, for smaller particles, using both aggregate or spherical representation results in a larger discrepancy when modelling the AAE or $MAC_{BC}$. The discrepancy was more pronounced in the cases of experiments E1, where the EC/TC analysis was not conducted. Therefore, the discrepancy could be a result of the presence of organic matter in smaller particles even after being heated by the Catalytic Stripper. The presence of larger percentage of organic matter in smaller particles is also observed from the results of the EC/TC analysis of experiment E2 where the largest $f_{oc}$ was observed in case VIII with the smallest $d_{p,V}$ of 86 nm ($d_{p,N}$ = 48 nm). These results in combination emphasize on the importance of morphology and size representation while modelling optical properties of BC particles.

**R2 GC7:** Maybe I missed it, but how was the SSA calculated? Using the CAPS or using the aethalometer + nephelometer combination, or the MAAP + nephelometer combination? How would the three calculations compare (after correcting for wavelength discrepancies?

**AR:** We thank the reviewer for the comment. The SSA was calculated using a combination of nephelometer and CAPS PM$_{ex\ 630}$. The absorption was also calculated using extinction minus scattering (ems), and compared to the MAAP and AE33. The extinction minus scattering (ems) shows better comparability to the MAAP.

[Figure]

The following line has been added in the Appendix A: Experimental setup and instrumentation:

The SSA was calculated using a combination of $\sigma_{scat}$ measured from nephelometer and extinction coefficient $\sigma_{ext}$ from the CAPS PM$_{ex\ 630}$.

The above figure is added in the supplementary for reference.

**R2 GC8:** In general, I found the paper a bit difficult and dry to read; most of the second paper is a list-like tedious description of the results in the figures with little commentary or explanation (potential or not) of the findings. While I don't have a lot of detailed suggestions on how to improve this issue, I think the authors could try to make the reading more fluent focusing more on commenting and interpreting the figures than on describing them. It would also help to use more consistent figures formatting and style and more readable (larger size and thickness) markers for example, although this last point, I understand, could be mostly a personal preference.

**AR:** We thank the reviewer for the comment. To add more clarity and enhance the flow and connectedness of the thoughts and expressions, we tried, whenever feasible, to include longer explanations and interpretations of the findings in the revised manuscript, as advised by the reviewer. Furthermore, in order to enhance the quality and presentation of the figures, the formatting and style have also been revised and improved.

**Specific comments (SC)**

Lines 58 – 61: black carbon aggregate compaction has also been associated with cloud (water and ice) processing alone (no organic coating).

**AR:** Thank you for the comment. The following line has been added after line 61 to include the information mentioned by the reviewer.

Additionally, cloud processing such as water condensation or evaporation also restructures the BC particles into more compact shapes (Bhandari et al., 2019).

Bhandari, J., China, S., Chandrakar, K. K., Kinney, G., Cantrell, W., Shaw, R. A., Mazzoleni, L. R., Girotto, G., Sharma, N., Gorkowski, K., Gilardoni, S., Decesari, S., Facchini, M. C., Zanca, N., Pavese, G., Esposito, F., Dubey, M. K., Aiken, A. C., Chakrabarty, R. K., Moosmüller, H., Onasch, T. B., Zaveri, R. A., Scarnato, B. V.,

Fialho, P. and Mazzoleni, C.: Extensive Soot Compaction by Cloud Processing from Laboratory and Field Observations, Scientific Reports, 9(1), 1–12, doi:10.1038/s41598-019-48143-y, 2019.

Line 77: Many others published on this issue, for example, the following paper might be relevant here: Scarnato, B. V.; Vahidinia, S.; Richard, D. T.; Kirchstetter, T. W., Effects of internal mixing and aggregate morphology on optical properties of black carbon using a discrete dipole approximation model. Atmospheric Chemistry and Physics 2013, 13, (10), 5089-5101.

**AR:** We thank the reviewer for sharing this important reference. It has been added in the revised manuscript.

To model the optical properties of such fractal BC aggregates, the Rayleigh-Debye-Gans (RDG) approximation (Sorensen, 2011), the discrete dipole approximation DDA (Purcell and Pennypacker, 1973), and the T-matrix method (Mackowski and Mishchenko, 1996) have been used (Adaichi et al., 2010; Kahnert, 2010; Li et al., 2016, Scarnato et al., 2013).

Scarnato, B. V.; Vahidinia, S.; Richard, D. T.; Kirchstetter, T. W., Effects of internal mixing and aggregate morphology on optical properties of black carbon using a discrete dipole approximation model. Atmospheric Chemistry and Physics 2013, 13, (10), 5089-5101.

Line 135: remove "the" in front of "both"

**AR:** Thank you for the correction. The change has been made in the revised manuscript.

Line 92: Many other papers before the one cited here discussed the nature of BC aggregates and monomers.

**AR:** We thank the reviewer for the comment. An earlier reference was also added in the revised manuscript.

Tian, K., Thomson, K., Liu, F., Snelling, D., Smallwood, G., and Wang, D.: Determination of the morphology of soot aggregates using the relative optical density method for the analysis of tem images, Combust. Flame, 144, 782–791, doi:10.1016/j.combustflame.2005.06.017, 2006.

Table 1: specify the wavelength of the SSA data in the table or caption itself.

**AR:** We thank the reviewer for the comment. It has been specified that the SSA was calculated at 660 nm in the Table 1.

Line 182: Many studies report the mixing configurations and morphologies of black carbon aggregates in the laboratory or ambient samples using scanning electron microscopy or transmission electron microscopy. How do the properties of the particles generated from the mini-CAST compare with those previous results?

**AR:** We thank the reviewer for highlighting this point. The results from TEM analysis have shown that BC particles emitted from a Mini-CAST burner have a fractal morphology. Depending upon the burning conditions of the flame described by the flame equivalence ratio, the percentage of organic matter which forms an external layer around the BC cores can vary (Mamakos et al., 2018). Ouf et al., 2016 showed that the structure of the BC aggregate depends on the operating conditions of the mini-CAST burner. They operated the mini-CAST burner on three conditions producing BC aggregates with $f_{oc}$ of 4% (CAST1), 47% (CAST2), and 87% (CAST3). Please see the TEM images from the three cases are shown for the reviewer comment GC3 above. The above discussion has already been incorporated in the revised manuscript as mentioned in GC 3.

Line 184: "kept in mind" is a bit vague.

**AR:** We thank the reviewer for the comment. The phrase "kept in mind" in changed to "considered" in the revised manuscript.

Figure 1: the mixed models all assume that the coating/mixing material did not affect the morphology of the bare aggregate underneath, correct? How good is that assumption? It is known that coating can cause very significant compaction of the initially bare black carbon particle.

**AR:** We thank the reviewer for the comment. Ouf et al. (2016) conducted NEXAFS analysis on BC produced from a diffusion flame-based mini-CAST burner and found that organics (by-products of the combustion) get attached to the edge of graphite crystallites without changing the inner structure of the core. This laboratory result can be simulated for coated BC in radiative modeling studies by assuming a spherical coating around each individual primary particle of a BC aggregate (Luo et al., 2018). This method was used to simulated coated fractal aggregates in our study. Both the composition and morphology of the aggregate play a role while choosing the representation for coated BC. The BC particles modelled in our study are assumed to have a less-compact and more chain-like structure with $f_{oc}$ up to 53%. In such cases, where the BC aggregate does not have a completely compact structure, the results are expected to be reliable (Luo et al., 2018). Moreover, Kahnert et al., 2017 compared the coating model (closed-cell model) used in this study to a realistic model, which showed good comparability.

The above explanation has been incorporated/modified in the Method section of the revised manuscript:

The "coated sphere" and "homogeneously mixed sphere" are the simplified models to represent coated BC aerosols. The "coated aggregate" is a realistic representation, morphologically similar to the "aggregate" (Figure 1b), with the difference that each monomer is coated with a layer of organic carbon. Ouf et al. (2016) conducted NEXAFS analysis on BC produced from a diffusion flame-based mini-CAST burner and found that organics (by-products of the combustion) get attached to the edge of graphite crystallites without changing the inner structure of the core. This laboratory result can be simulated for coated BC in radiative modeling studies by assuming a spherical coating around each individual primary particle of a BC aggregate (Luo et al., 2018). This method was used to simulated coated fractal aggregates in our study. Both the composition and morphology of the aggregate play a role while choosing the representation for coated BC. The BC particles modelled in our study are assumed to have a less-compact and more chain-like structure with $f_{oc}$ up to 53%. In such cases, where the BC aggregate does not have a completely compact structure, the results are expected to be reliable (Luo et al., 2018). Moreover, Kahnert et al., 2017 compared the coating model (closed-cell model) used in this study to a realistic model, which showed good comparability.

Ouf, F. X., Parent, P., Laffon, C., Marhaba, I., Ferry, D., Marcillaud, B., Antonsson, E., Benkoula, S., Liu, X. J., Nicolas, C., Robert, E., Patanen, M., Barreda, F. A., Sublemontier, O., Coppalle, A., Yon, J., Miserque, F., Mostefaoui, T., Regier, T. Z., Mitchell, J. B. A. and Miron, C.: First in-flight synchrotron X-ray absorption and photoemission study of carbon soot nanoparticles, Scientific Reports, 6, doi:10.1038/srep36495, 2016.

Luo, J., Zhang, Y., Zhang, Q., Wang, F., Liu, J., and Wang, J.: Sensitivity analysis of morphology on optical properties of soot aerosols, Opt. Express, 26, A420–A432, https://doi.org/10.1364/oe.26.00a420, 2018b.

Kahnert, M.: Optical properties of black carbon aerosols encapsulated in a shell of sulfate: comparison of the closed cell model with a coated aggregate model, Opt. Express, 25, 24579–24593 https://doi.org/10.1364/oe.25.024579, 2017.

Line 193 "Leaving some residuals" how much is "some" can the authors be more quantitative?

**AR:** We thank the reviewer for this comment. The uncertainty associated with the Catalytic Stripper in removing the organic matter was studied by Mamakos et al., 2013. It has been reported that in the 21–250∘C temperature range, the Catalytic Stripper is able to remove up to 96% of the more volatile fraction of organic matter. However, in the 250–500∘C temperature range, the Catalytic Stripper removes 30–60% of the less volatile organic matter. In our study, we model the optical properties for particles passed through the Catalytic Stripper at 350∘C, therefore, expecting 40-70% of the less volatile organic matter residues.

The above points shall be summarized in the methods section in the revised manuscript as follows:

For modelling the particles from the denuding experiment E1, the simulated particles are assumed to be bare black carbon, since a Catalytic Stripper was used to remove the volatile organic matter. Some residuals, however, are left behind by the Catalytic Strippers, depending on the volatility of the organic matter. Mamakos et al. (2013) reported that in the 21–250∘C temperature range, the Catalytic Stripper is able to remove up to 96% of the more volatile fraction of organic matter. However, in the 250–500∘C temperature range, the Catalytic Stripper removes 30–60% of the less volatile organic matter. This is noteworthy when comparing the modelled optical results with their equivalent laboratory measurements.

Line 225: How did the authors deal with issues induced by non-sphericity and multiple charges in the SMPS measurements?

**AR:** We thank the reviewer for this comment. The SMPS software used in this study for correcting the multiple charges did not account of non-sphericity of the aerosols. Gulijk et al., 2003 reported that the problem of multiple charges is mainly due to the fact that a substantial number of fractal-like particles larger than 1000 nm mobility diameter are present in a diesel soot aerosol. In contrast with diesel soot, PNSDs for BC particles generated with mini-CAST generators during experiments E1 and E2 show significantly fewer particles larger than a mobility diameter of 500 nm. In the future, it would be ideal to have an SMPS software that accounts for multiple charges in non-spherical particles.

Van Gulijk, C., Marijnissen, J. C. M., Makkee, M., Moulijn, J. A. and Schmidt-Ott, A.: Measuring diesel soot with a scanning mobility particle sizer and an electrical low-pressure impactor: Performance assessment with a model for fractal-like agglomerates, Journal of Aerosol Science, 35(5), 633–655, doi:10.1016/j.jaerosci.2003.11.004, 2004.

Line 231: this approach assumes that the coating thickness is the same over each particle in the particle ensemble, correct? However, it has been shown that the amount of coating can be quite variable from particle to particle within an ensemble of ambient particles. Particle-resolved models suggest that this coating distribution can also induce significant deviations in simulated optical properties.

**AR:** Thank you for the comment. We agree with the reviewer that for BC particles in the atmosphere, the amount of coating can vary in distribution over the entire particle. However, the TEM images of BC produced from mini-CAST soot generators did not show any significant heterogeneity in the distribution of the coating (Ouf et al., 2016; Ess et al., 2021). In order to assume a variable coating around the BC particle, extensive TEM images should be produced for simulating such particles. As a result, using a uniform coating can be a simple and generic method for coating BC particles. Moreover, Kahnert et al., 2017 compared the coating model (closed-cell model) used in this study to a realistic model, which showed good comparability.

The above explanation has been incorporated/modified in the Method section of the revised manuscript:

Ouf et al. (2016) conducted NEXAFS analysis on BC produced from a diffusion flame-based mini-CAST burner and found that organics (by-products of the combustion) get attached to the edge of graphite crystallites without changing the inner structure of the core. This laboratory result can be simulated for coated BC in radiative modeling studies by assuming a spherical coating around each individual primary particle of a BC aggregate (Luo et al., 2018). This method was used to simulated coated fractal aggregates in our study. Both the composition and morphology of the aggregate play a role while choosing the representation for coated BC. The BC particles modelled in our study are assumed to have a less-compact and more chain-like structure with $f_{oc}$ up to 53%. In such cases, where the BC aggregate does not have a completely compact structure, the results are expected to be reliable (Luo et al., 2018). Moreover, Kahnert et al., 2017 compared the coating model (closed-cell model) used in this study to a realistic model, which showed good comparability.

Line 248: It seems like this "first method" uses mass information, but in line 247, the authors mention that they did not use mass information (?).

**AR:** In previous studies comparing modelled and measured optical properties of soot aggregates, the $N_{pp}$ was determined by dividing the measured mass of total particle by the estimated mass of a spherule (Forestierti et al., 2018). In our study, we could not measure the mass of a single particle. Therefore, in the first method to estimate $N_{pp}$, the particle mass is estimated using the $\rho_{eff}$ (Rissler et al. 2013) from the measured particle diameter.

The above explanation has been incorporated/modified in the revised manuscript as shown in red text below:

In the previous studies about the comparison of modelled and measured optical properties of soot aggregates, the $N_{pp}$ was determined by dividing the measured mass of total particle by the estimated mass of a spherule (Forestierti et al., 2018); or reconstructed using results from TEM analysis (He et al., 2015). In our study, we investigated the methods for estimating the $N_{pp}$ in absence of mass or TEM results. Three different conversion methods for calculating the number of primary particles $N_{pp}$ per aggregate were applied in this study. In the first method by Rissler et al. 2012, the particle mass estimated using the $\rho_{eff}$ is divided by the estimated mass of a spherule.

Line 255: why 14 nm? And how good is this assumption? Does it matter?

**AR:** We thank the reviewer for this comment. Diffusion flame-based generators like the mini-CAST burners, produce soot primary particle radius ($a_{pp}$) between 4 and 14 nm (Mamakos et al., 2013; Bourrous et al., 2018). Kahnert (2010) pointed out the insensitivity in the optical properties when the radii of the primary particle fall in the range of 10–25 nm. In the sensitivity study of $a_{pp}$ given in section 3.2.3, it is shown that for an $a_{pp}$ value between 10 and 14 nm, the modelled SSA is in good agreement with the measured SSA. In our study, we therefore choose the upper limit of 14 nm.

The above explanation has been incorporated/modified in the revised manuscript as indicated below in red text:

Kahnert (2010) pointed out the insensitivity in the optical properties when the radii of the primary particle fall in the range of 10–25 nm. Due to absence of measurements of $a_{pp}$, and for the sake of simplicity, a constant value $a_{pp} = 14$ nm was used for the entire study, except for the part of sensitivity analysis discussed in the next section.

Lines 263 – 265: These two sentences are not clear to me.

**AR:** Before simulating the BC fractal aggregates using the DLA software, the outer radius of the primary particle ($a_o$), and the inner radius of the primary particle ($a_{in}$) following equation (2). The two sentences mean to clarify that the diffusion limited aggregation software generates the BC aggregate using $a_o$, and not $a_{in}$.

The above has been modified for clarity in the revised manuscript as follows:

Following equation (2), using the fraction of organic carbon ($f_{oc}$), the outer radius of the primary particle ($a_o$) and the inner radius of the primary particle ($a_{in}$) were determined. It must be noted that in the "coated aggregate" representation, the primary particles of the aggregates generated from the Diffusion Limited Aggregation (DLA) software have a radius equal to $a_o$. In the next step, a smaller sphere with a radius of $a_{in}$ is placed at the center of the primary particle representing the BC core.

Lin 433: Maybe "in contrast" or "on the contrary" instead of "in contrary"?

**AR:** Thank you for the correction. We changed "in contrary" to "on the contrary" in the revised manuscript.

Section 3.1.1 I am not sure I fully understand the rationale for calculating Npp from dp,V vs. dp,N, all the methods listed in appendix B seem to be using dp,N

**AR:** Thank you for the comment. The equations in appendix B are written in terms of the number number mean mobility diameter ($d_{p,\bar{N}}$). However, for the sake of comparison, the $N_{pp}$ is calculated using both number mean mobility diameter ($d_{p,\bar{N}}$) and the volume mean mobility diameter ($d_{p,\bar{V}}$). This was done in order to make a comparison how the optical properties differ when the aggregate is generated using $d_{p,\bar{N}}$ and $d_{p,\bar{V}}$.

The above point has been summarized and mentioned in line 845 of the revised manuscript:

For all the three conversion methods, the $N_{pp}$ was estimated using both the number mean mobility diameter ($d_{p,\bar{N}}$), and the volume mean mobility diameter ($d_{p,\bar{V}}$).

Line 449: missing "to" in front of "the experimentally…"

**AR:** Thank you for the correction. We added "to" in the revised manuscript.

Line 453: There are also numerical studies that show the effect of measured compaction on scattering.

**AR:** We thank the reviewer for this comment. The following numerical and measurement studies have been added in the revised manuscript.

Luo, J., Zhang, Y. and Zhang, Q.: The Ångström exponent and single-scattering albedo of black carbon: Effects of different coating materials, Atmosphere, 11(10), 1103, doi:10.3390/atmos11101103, 2020.

Yuan, C., Zheng, J., Ma, Y., Jiang, Y., Li, Y. and Wang, Z.: Significant restructuring and light absorption enhancement of black carbon particles by ammonium nitrate coating, Environmental Pollution, 262, 114172, doi:10.1016/j.envpol.2020.114172, 2020.

Figures 5 and 6: I think it would be nice to have these figures be the same for both methods, meaning showing SSA and AAE in both, or maybe even better, showing SSA, AAE, and sigma abs in both.

**AR:** We thank the reviewer for this comment. As suggested, the $\sigma_{abs}$ will be shown for the monodisperse method, and he AAE will be shown for the polydisperse method.

[Figure]

Line 544: "the" in front of "nature"? In general, this sentence is a bit unclear to me.

**AR:** We thank the reviewer for the correction. The sentence has been explained more in the revised version of the manuscript:

Moreover, the nature of the dependence between the refractive index and optical properties differed depending on whether spherical or aggregate representations were used.

Line 447: "the modelled results could not be validated with the modelled findings" maybe the authors mean "the modelled results could not be validated with measurements"?

**AR:** Thank you for the correction. It has been changed to "the modelled results could not be validated with measurements" in the revised manuscript.

Line 660: similarly, "modelled" or "measured"?

**AR:** Thank you for the correction. It has been change to "measured".

Line 689: "which pronounces" reads a bit awkward in the sense that the sentence is not very clear on what "which" refers to.

**AR:** Thank you for the comment. The sentence has been made more clear in the revised version of the manuscript as follows:

In the case of monodisperse particles, the modelled SSA using "sphere" representation was higher than the measured SSA, and the difference grew further as the particle size increased.

Appendix A: the authors mentioned the use of an Aurora4000 nephelometer, if I am not mistaken, that instrument is a polar nephelometer that should allow estimating the asymmetry parameter g using the backscatter fraction (e.g., Moosmüller, H.; Ogren, J. A., Parameterization of the Aerosol Upscatter Fraction as Function of the Backscatter Fraction and Their Relationships to the Asymmetry Parameter for Radiative Transfer Calculations. Atmosphere 2017, 8, (8), 133.). Therefore, I am unclear why the authors did not attempt comparing g measured with g simulated in figure 11.

**AR:** Thank you for the comment. Estimating asymmetry parameter *g* from nephelometers is very uncertain. For e.g. the simple parameterizations using the hemispheric backscatter fraction (Andrews et al., 2006) were derived for ambient and more spherical aerosol particles. However, it is not clear how this parameterization works for BC aerosols with low single scattering and fractal morphology. The limitations of the Aurora 4000 nephelometer (Müller et al., 2012) is that the polar function is measured in up to 18 angular sectors in forward scattering direction, whereas the real resolution is smaller, since the shutter function is not steep enough (Figure below from Müller et al., 2012). Furthermore the hemispheric backscattering is just represented as one large sector (scattering angle 90 to 180°). There is still a need to examine Aurora4000 in more depth to determine an asymmetry parameter for fractal soot particles.

[Figure]

Illumination function of the Aurora4000.

Andrews, E., Sheridan, P. J., Fiebig, M., McComiskey, A., Ogren, J. A., Arnott, P., Covert, D., Elleman, R., Gasparini, R., Collins, D., Jonsson, H., Schmid, B., and Wang, J.: Comparison of methods for deriving aerosol asymmetry parameter. J. Geophys. Res.- Atmos., 111, D05S04, doi:10.1029/2004JD005734, 2006.

Müller, T., Paixão, M., Pfeifer, S., and Wiedensohler, A.: Scattering Coefficients and Asymmetry Parameters derived from the Polar Nephelometer Aurora 4000, in: EEuropean Aerosol Conference EAC 2012, Granada, 2–7 September 2012, Zenodo [poster],https://doi.org/10.5281/zenodo.5588445, 2012